# A Sustainable Method for Publishing Interoperable Open Data on the Web

**Raf Buyle [1,*], Brecht Van de Vyvere [1], Julián Rojas Meléndez [1], Dwight Van Lancker [1,2], Eveline Vlassenroot [3], Mathias Van Compernolle [3], Stefan Lefever [4], Pieter Colpaert [1], Peter Mechant [3] and Erik Mannens [1]**

1   Imec—IDLab, Ghent University, 9052 Gent, Belgium; brecht.vandevyvere@ugent.be (B.V.d.V.); julianandres.rojasmelendez@ugent.be (J.R.M.); dwight.vanlancker@ugent.be (D.V.L.); pieter.colpaert@ugent.be (P.C.); erik.mannens@ugent.be (E.M.)
2   Digitaal Vlaanderen, Flemish Government, 9052 Zwijnaarde, Belgium
3   Imec—MICT, Ghent University, 9000 Ghent, Belgium; eveline.vlassenroot@ugent.be (E.V.); mathias.vancompernolle@ugent.be (M.V.C.); peter.mechant@ugent.be (P.M.)
4   Imec EdiT, 2000 Antwerp, Belgium; Stefan.Lefever@imec.be
*   Correspondence: raf.buyle@ugent.be; Tel.: +32-478-47-61-37

**Abstract:** Smart cities need (sensor) data for better decision-making. However, while there are vast amounts of data available about and from cities, an intermediary is needed that connects and interprets (sensor) data on a Web-scale. Today, governments in Europe are struggling to publish open data in a sustainable, predictable and cost-effective way. Our research question considers what methods for publishing Linked Open Data time series, in particular air quality data, are suitable in a sustainable and cost-effective way. Furthermore, we demonstrate the cross-domain applicability of our data publishing approach through a different use case on railway infrastructure—Linked Open Data. Based on scenarios co-created with various governmental stakeholders, we researched methods to promote data interoperability, scalability and flexibility. The results show that applying a Linked Data Fragments-based approach on public endpoints for air quality and railway infrastructure data, lowers the cost of publishing and increases availability due to better Web caching strategies.

**Keywords:** smart cities; IoT; semantic web; Linked Open Data; air quality; railway infrastructure; Linked Data Fragments

## 1. Introduction

Today, most of the global population lives in urban areas, and it is expected that this will increase to nearly 70% by 2050 [1]. Such high levels of population growth create problems in waste management, air pollution and traffic mobility [1,2]. To avoid this accelerated urbanization turning into a crisis, cities must become "smart." "Smart" refers to a continuous comprehensive commitment to innovation in technology, management and policy [3].

To become smart and make smarter decisions, cities need to amalgamate disparate data sources, including data on urbanization, weather and traffic [4]. The resulting insights can support better policymaking, such as better urban planning decisions on where to build new roads, schools or hospitals. For example, providing real-time air quality data to citizens can help them choose a route that reduces their exposure to pollution [5,6]. However, governments are struggling to create and maintain accessible public endpoints where high-value datasets can be published because of barriers such as availability, scalability and publishing costs [7].

In the State of the Union at the European Parliament Plenary, President von der Leyen addressed the need for common data spaces to ensure data are widely accessible [8]. Meeting the challenges to ensure that data are accessible and interoperable is at the core of the European strategy to create a single market for data [9]. According to the European

data strategy, interoperable data spaces can bring extensive benefits in domains such as the European Green Deal data space and the Common European Mobility data space [9]. For smart cities, the need to publish interoperable data is crucial in domains such as sustainable energy [10], water supply networks management [11], or policymaking [12,13].

Thus, the key question is how public authorities can develop a sustainable method for publishing open data. A key consideration is how the cost between data publishers and consumers should be distributed.

In this article, we assess the cost-efficient publishing of sensor data time series on air quality, which could help reach the ambitious European target to become the first climate-neutral continent by 2050 [9]. Additionally, to assess the cross-domain applicability of our data publishing approach, we explore the cost-efficient open data publishing of European railway infrastructure information, based on the data resources published by the European Agency for Railways (https://www.era.europa.eu/, accessed on 26 July 2021) (ERA).

Despite originating from different domains and dealing with different types of data, both use cases present good examples, within the European context, of the challenges faced by public authorities acting as open data providers. Our approach builds upon the same architectural building blocks to provide a cost-efficient and cross-domain open data publishing alternative.

Our research on the first use case assumes that the application of Linked Data Fragments on public endpoints will result in more efficient caching for air quality sensor data, lowering the cost of publishing and increasing data availability. Furthermore, this approach assumes that Linked Data Fragments will support the business needs for an air quality sensor data endpoint. Our research is somewhat limited as only one dataset of sensor data is used, relying only on geotemporal query operations, similar to the Next Generation Service Interfaces (NGSI) specification [14]. In a real-world application, more context, and thus datasets, will need to be queried.

The second use case concerns a typical scenario for railway infrastructure data, using the calculation of possible routes across the railway network and, in particular, a route compatibility check. This use case recognises the need to support potentially complex geospatial query operations that perform graph shortest-path algorithm calculations. Traditional solution systems typically rely on dedicated database systems that support geospatial querying, such as PostGIS3, or graph databases such as Neo4J [15,16]. However, these systems result in heavy computational costs on open data publishers, which often need to impose query limitations to guarantee service availability.

This article is made up of six sections. Section 2 discusses related work. First, we introduce smart cities and the two use cases. Next, we discuss the nature of air quality sensor data and explain why we need a different layer where sensor data can be connected and interpreted by machines on a Web-scale. We outline the design principles of Linked Data, apply them to sensor data and elaborate on railway infrastructure data [17]. Finally, we describe caching strategies to publish data and assess how the roles of data publishers and data consumers can be balanced. Section 3 describes how the use case scenarios can benefit from a Linked Data Fragments approach. Section 4 benchmarks the FIWARE (https://www.fiware.org/about-us/, accessed on 26 July 2021) and Linked Data Fragments architectures for the air quality sensor data use case and assesses the cost for both the data publisher and the consumer. Section 5 discusses the findings of the benchmark in detail, which should help government agencies and organizations to re-use the architectural components when refactoring the endpoints. We also discuss the cross-domain interoperability and architectural flexibility applied to the use case of a route compatibility check. We point out methods that lower the cost for publishing and increase the availability of endpoints and the flexibility of client-side applications. We show how governments can distribute the cost between the data publisher and the consumer and point out how these insights can lead to a sustainable sensor network promoting interoperability, flexibility, availability, scalability and predictability.

## 2. Background and Related Work

### 2.1. Smart Cities, Air Quality Sensor Data and Railway Infrastructure Data

"Smart" cities are not a new phenomenon. The ancient city of Rome accommodated between five hundred thousand and one million inhabitants through an advanced bureaucratic information system and efficient waste management [18]. Similarly, air pollution was monitored in ancient times and recorded in poems (e.g., by Horace (65 BC–8 AD)) [19]. The first regulations on air quality can be observed in Roman Law: "Aerem corrumpere non licet" (Air pollution is not allowed) [20]. Today, according to the World Health Organization (https://www.who.int/, accessed on 26 July 2021) (WHO) "air pollution represents the biggest environmental risk to health" [21]. The WHO estimates that outdoor air pollution caused three million deaths in 2012. Urbanisation has increased the concentration of ambient air pollution resulting from traffic, transport, domestic heating and industrial emissions [22]. Air quality in Europe is regulated by Directive 2008/50/EC, which defines the threshold for the concentration of several pollutants.

The European Commission [23] is fostering the re-use of high-value datasets, such as sensor data time series on air quality, by legislating for (real-time) datasets to be published in a machine-readable format and to be automatically transferable through an Application Programming Interface (API). An API allows users to query and combine data from several endpoints, without maintaining copies of the data. Machine-readable data enables information to become self-describing removing the need for manual analysis and transformations.

Since public city administrations cannot predict the load caused by users of any given dataset on their APIs, services often lack elasticity. An example is the launch of the "solar map" (https://apps.energiesparen.be/zonnekaart, accessed on 26 July 2021) in Flanders (the northern part of Belgium). The "solar map" is an online application provided by the Flemish Energy Agency (https://www.energiesparen.be/over_veka, accessed on 26 July 2021) that shows the suitability of solar panels for any given roof and calculates the payback period for investing in solar panels. The application builds on remote sensing data from the Digital Flanders Agency. During the public launch, the application went down because the services were under dimensioned [24]. These problems will be exacerbated as the Web of Sensors becomes a distributed, high-volume, high-velocity and heterogeneous mix of sensor and storage platforms [25].

Within the context of mobility, the European Commission states that "real-time notification of delayed trains can save 27 million working hours. This amounts to €740 million in labour costs" [9]. For this reason, and motivated to foster safe and efficient railway transportation operations, the European Agency for Railways (ERA) manages and publishes different base registries (some of them as open data) [12,26,27] containing valuable information related to the European railway infrastructure. These data may be re-used by the different stakeholders of the railway domain to improve their services through innovative data powered applications. Similar to public government agencies in Flanders, the ERA also faces the challenges of making their public data accessible, interoperable and available for maximum re-use.

### 2.2. Air Quality Sensor Data

#### 2.2.1. (Sensor) Data Streams

Air quality in Europe is regulated by Directive 2008/50/EC, which defines the threshold of the concentration of several pollutants, including fine particles (PM2.5), Sulphur dioxide ($SO_2$), Nitrogen dioxide ($NO_2$), PM10 and Carbon monoxide (CO) [28]. These pollutants are measured using Air Quality sensors. Good quality sensors used to be costly and were based on chemical analyses and, thus, less densely deployed [22]. Recently, low-cost sensors, including electrochemical sensors ($NO_2$, CO, $SO_2$ gas detection) and optical particle sensors for PM10, entered the mainstream market at affordable prices [22]. A disadvantage of these sensors is that they must be calibrated due to their inferior me-

chanical and electrical tolerances. Furthermore, their signal often changes independently of the measurements due to sensor drift [29].

To achieve a denser sensor-grid, low-cost sensors can be deployed at high volumes. The data can be calibrated using the data of the less-dense high-end air quality sensors and can be combined with other valuable datasets, including weather and traffic.

To achieve this, we need a "layer" where sensor data can be connected and interpreted by machines. This layer will enable the blanks to be filled out by interpolating nearby and historical data. The results can then be "re-gridded" to fit a uniform "time-space" dataset, which can then be easily analysed.

Nittel [30] defines a sensor data stream as "a time series of sensor measurements $m_{sj} = < t_i, l_{sj}, v_1; v_2,...v_n >$ generated by a sensor node $s_j$, based on one or more of its attached sensors". Both the timestamp $t_i$ and the location of the sensor $l_{sj}$ are crucial to interpret the sensed value $v_n$. The location can be a fixed value, in the case of the high-end stationary air quality sensors, or a variable value, derived from, e.g., Global Positioning System (GPS), where the low-cost sensors are mounted on a vehicle. Furthermore, information related to the type (e.g., PM10, NO2), the calibration parameters (including relative humidity and temperature) and the quality of the measurement is important [31]. It is expected that sensor streams will evolve to become spatially densely distributed with a high-frequency sampling rate [30] generating high volumes of data. Considering that there are four thousand air quality sensors and a sample each second, this represents over 126 billion samples collected per year, compared to another high-volume datasets in the financial sector; this exceeds the number of bank transactions in Europe in 2018 [32].

### 2.2.2. Interoperability of Air Quality Sensor Data

When considering air quality sensor data, an important challenge is to identify and process the heterogeneous and mixed quality datasets (Hendler, 2014). Therefore, interoperability (IOP) is crucial, both for combining air quality data from different sources as well as for linking these data to other datasets such as traffic or weather data [33]. The European Commission defines IOP as the ability of organisations to share information and knowledge, through the business processes they support, by exchanging data between their ICT systems [26]. To ensure that sensor data can be re-used, various IOP levels should be addressed in turn; namely the legal, organisational, technical and semantic level [27], see Table 1.

**Table 1.** Northbound Air Quality Sensor Data interoperability levels evaluated using the European Interoperability Framework [26].

| | |
|---|---|
| Legal Interoperability | Non-interoperable legislation, data licenses for maximum re-use. |
| Organisational Interoperability | Aligned and documented business processes, Service Level Agreements and appropriate archiving mechanisms for streaming data. |
| Technical Interoperability | HTTP as the foundation for data communication and URIs to identify "things". |
| Semantic Interoperability | Information is aligned on standardised vocabularies. The method of Linked Data facilitates semantic and syntactic IOP. |

As IOP frameworks—including the European Interoperability Framework—assume a hierarchy in the IOP levels, legal and organisational IOP can only be implemented successfully when semantic and technical IOP are in place [34].

First, as smart cities are networked ecosystems, organisations broaden their activities outside their policy domain, which results in legislative barriers, introduces costs and slows down innovation [35,36]. These barriers impede legal IOP and originate because of (a) non-interoperable legislation between different governmental levels such as municipalities and regional government, (b) non-interoperable laws across different policy domains such

as environmental regulations and mobility and lastly, (c) (the lack of) clauses in agreements between governments and software vendors prohibiting the re-use of data.

Second, to create a sustainable sensor network, business processes among actors in the ecosystem must be aligned and documented, for instance by requiring service providers to agree on a Service Level Agreement framework [27,37]. These efforts on coordinated business processes, responsibilities and expectations are referred to as efforts towards organisational IOP [26].

Third, technical IOP covers the interconnection of applications and infrastructures, including interface specifications that interconnect systems and services [26]. In the Internet of Things (IoT) paradigm, objects that both harvest information from the physical world (sensors) and interact with their environment (actuators) are interconnected [38]. In these networks, we distinguish northbound (NBI) and southbound interfaces (SBI).

An SBI provides connectivity to the low-level components in the physical infrastructure such as sensors and actuators. Alternatively, an NBI provides connectivity with the other network nodes, regularly exposed as APIs. These APIs can shield the disparateness of the physical infrastructure and create a heterogeneous NBI, reducing the complexity of application development [39].

Sensors will not only generate an excessive amount of data but more importantly, data of a greater variety [38,40]. Hendler [41] defines "broad data" as the phenomenon of "trying to make sense out of a world that depends increasingly on finding data that is outside the user's control, increasingly heterogeneous, and of mixed quality". To face the challenges of broad IoT data, the principles of Linked Data enable data to become self-describing and machine-readable [40,42]. Machine-readable data allow autonomous agents to reason on the sensor data [43]. Linked Data build upon the Web and use typed links between data entities from disparate sources [17]. These links are typed statements, described using the Resource Description Framework [44]. The entities are globally unique and identified using Uniform Resource Identifiers (URIs). URIs can be consulted using the HyperText Transfer Protocol (HTTP), which is the foundation for data communication on the Web [45,46].

Finally, a lack of semantic agreements causes multiple transformations on the different data models and syntaxes, which implies rewiring APIs and induces exorbitant costs [33]. The European Interoperability Framework (EIF) refers to semantic IOP as the meaning of information that is preserved and understood during the exchange between all communicating parties [27] and the purpose of this level is that it "encompasses the intended meaning of the concepts in the data schema" [47]. In this way, semantic IOP can tackle heterogeneity across datasets and ensure that no different terms are used for a given attribute or that a given term is not used to represent different concepts [48]. It includes both semantic interoperability, which refers to the meaning of the sensor data, and syntactic interoperability, which specifies the grammar of the information such as XML or JSON [26]. The competing vocabularies that model the domain of air quality from slightly different viewpoints, including INSPIRE, NGSI-LD and SSN/SOSA, are discussed in Section 2.3 (see Table 2).

**Table 2.** Overview of the characteristics of three ubiquitous vocabularies [14,49–51].

| Vocabulary | Use for | Wide Industry/Community Support | Ratified Vocabulary for Air Quality | Linked Data Support |
| --- | --- | --- | --- | --- |
| INSPIRE | INSPIRE | NO | YES | NO |
| NGSI-LD (FIWARE Data model "Air quality observed) | ETSI | YES | YES (https://github.com/smart-data-models/dataModel.Environment/tree/master/AirQualityObserved, accessed on 26 July 2021) | YES |
| SSN/SOSA | W3C/OGC | YES | YES | YES |

### 2.2.3. Semantic IOP Applied to Air Quality Data

Space, time and theme are key dimensions for registering and analysing sensor data, as they make it possible to link the sensor data to other datasets [25]. The spatial component provides information about the location, the temporal attributes observe the time and time-zone, while the thematic attributes provide information about the sensor type [52]. In the context of this article, we will focus on the air quality data that monitor several pollutants, including fine particles and Nitrogen dioxide.

European Member States are obliged by law to report ambient concentrations; when thresholds are exceeded, they need to inform the public [53]. The European Commission established a legal framework, "Infrastructure for Spatial Information in the European Community" (INSPIRE), that focuses on accessible and interoperable data [54]. INSPIRE defines data specifications and implementing guidelines for exchanging air quality data, including a standardised description for sensors, sensor location, orientation, as well as the sensor's geometric, dynamic and radiometric characteristics [51]. The conceptual schemas, which make up the normative part of the standard, are defined in the Unified Modelling Language (UML) and in XMLschema. XMLschema is a description of a type of Extensible Markup Language (https://www.w3.org/XML/, accessed on 26 July 2021) (XML) document that defines a set of rules for encoding documents in a format that is both human-readable and machine-readable.

In 2016, the European Commission (EC) requested that the European Telecommunications Standards Institute (ETSI) (https://www.etsi.org/, accessed on 26 July 2021) create an Industry Specification Group (ISG) to define a standardised API for Context Information Management (CIM) with Future Internet Ware (FIWARE) Next Generation Service Interfaces (NGSI) as a nominee. FIWARE is an open-source platform, supported by the EC. NGSI is a protocol to manage Context Information. The ISG delivered the Next Generation Service Interfaces as a Linked Data (NGSI-LD) standard [14], which enables nearly real-time access to information from different distributed data sources. The NGSI-LD Information Model Structure (IMS) consists of the following two layers: a core Meta-model and a Cross-Domain Ontology that can be extended with domain-specific logic. The core Meta-model defines a minimal set of constructs that are the basic building blocks of the Cross-Domain Ontology including Entity, Relationship, Property and Value [14,55]. The Cross-Domain Ontology describes concepts and constraints that provide consistency between the different IoT domains and applications; these concepts include Geographical properties, Temporal properties and Time values [14,55]. The domain-specific logic can be extended with ontologies for a specific domain, including air quality, noise level and water quality (https://github.com/FIWARE/data-models/blob/master/specs/ngsi-ld_howto.md, accessed on 26 July 2021). NGSI-LD requires a reimplementation of existing Linked Data domain models to fit the semantics of NGSI.

In 2017, the World Wide Web Consortium (https://www.w3.org/, accessed on 26 July 2021) (W3C) and the Open Geospatial Consortium (OGC) Spatial Data on the Web (https://www.w3.org/2017/sdwig/, accessed on 26 July 2021) (SDW) working group joined forces and developed a set of ontologies that annotate sensors, actuators, samplers and their time series [49,50]. The ontologies include a lightweight core Sensor, Observation, Sample and Actuator (http://www.w3.org/ns/sosa/, accessed on 26 July 2021) (SOSA) ontology and the more expressive Semantic Sensor Network Ontology (http://www.w3.org/ns/ssn/, accessed on 26 July 2021) (SSN) [49]. As such, SOSA provides a minimal core for SSN and ensures minimal IOP. According to Haller, SSN and SOSA support various use cases including "satellite imagery, large-scale scientific monitoring, industrial and household infrastructures, social sensing, citizen science, observation-driven ontology engineering, and the Web of Things" [56]. The SSN and SOSA ontologies are available in line with the principles of Linked Data, which allow autonomous agents to reason on the capabilities, measurements and provenance of an individual sensor or a network of sensors. Finally, there is the ISO/OGC Observations and Measurements (O&M) standard to model air quality data. This model is used in the OGC's SensorThings API, a REST-based API for

sensing and tasking IoT devices. SensorThings uses a similar service-oriented approach as NGSI-LD where, among others, filtering capabilities are offered to a client.

### 2.3. Interoperability Applied to Railway Infrastructure Data

Since railway infrastructure data are a networked ecosystem where various infrastructure managers—such as Infrabel in Belgium—interact and need to exchange information, interoperability is crucial. The different IoP levels are documented in the implementing regulation 2019/773 [57] of the European Commission. On a legal level IoP, the European Commission defined regulation relating to the operation and traffic management of the rail system within the European Union [57].

To create an interoperable rail network, rules and procedures among actors in the ecosystem are aligned and documented such as the ability to exchange information about a train arriving, passing or departing from a station [57].

In the railway domain, most efforts have focused on providing concise and reusable definitions for the different concepts and elements that make up the railway infrastructure. A common data model would enable an automated exchange of railway infrastructure data among the different stakeholders (e.g., infrastructure managers and railway vehicle operators) to support safe service operations. Currently, several semantic and non-sematic data models exist, all sharing the goal of increasing data interoperability in the railway domain.

Among the non-semantic models, we can find RailML, which defines an XML Schema covering multiple aspects of the railway infrastructure [58]. RailML, re-uses the meta model for topological entities defined by the RailTopoModel, which aims to bridge the different modelling approaches with a unique graph-based logical model of the railway network topology [59].

Similarly, Semantic Web-driven approaches exist to provide a framework for data interoperability in the railway domain. For instance, Tutcher applies a structured methodology to create an OWL-based ontology for the railway domain. Verstichel et al. [60] define a semantic model and discuss how semantic technologies may support data integration in the railway domain. Bischof et al. take advantage of the domain knowledge embedded in established models such as RailML to derive a reusable ontology [61]. Most recently, ERA published an initial sematic vocabulary (http://era.ilabt.imec.be/era-vocabulary/index-en.html, accessed on 26 July 2021) to model railway topology networks, together with other relevant domain concepts.

However, reliable railway infrastructure open data are not particularly easy to find. Crowd-sourced initiatives such as OpenStreetMap (OSM) provide a rich source of open data related to the railway domain [62]. For instance, the OpenRailwayMap (https://www.openrailwaymap.org/, accessed on 26 July 2021) is a data visualization service built on top of OSM data. However, these data are not always directly reusable. An example of authoritative data can be found in the open data portal of the Belgian infrastructure manager Infrabel, which includes data dumps of the Belgian railway infrastructure. Yet, these data dumps lack formal semantic definitions and are not immediately queryable and reusable by applications (https://opendata.infrabel.be/explore/?sort=explore.popularity_score&refine.theme=Infrastructure&disjunctive.keyword, accessed on 26 July 2021).

### 2.4. Data Caching Strategy

As the Sensor Web is distributed, multimodal (e.g., air quality, relative humidity, temperature, reference data), read-intensive and subject to large-scale load variations, it becomes very brittle [25]. To lower a server's central processing unit (CPU) load—and thus the actual publishing cost—optimisations can be implemented via caching, which reduces traffic [63,64]. Caching stores data, which lowers the cost of handling future requests. In this section, we explain how Web caching—or HTTP caching—can reduce the need for client-server interaction.

The World Wide Web (WWW) has a software architecture that is designed for internet-scale across organisational boundaries and builds upon the principles of a distributed hypermedia application. To raise scalability, Web applications follow the Representational State Transfer (REST) architectural style—which is demarcated by a set of architectural constraints that enable caching—and is a blueprint for the behaviour of a well-designed Web application. The REST architectural style resembles the human Web, which builds upon hyperlinks, and a set of architectural constraints that facilitate architectural elasticity [65]. The three most essential constraints are (i) uniform interface, (ii) client-server and (iii) stateless and cache constraints.

First, the uniform interface simplifies the architecture and empowers clients to evolve separately [66]. This key feature of the REST architecture unfolds in four sub-constraints; namely (a) Uniform Resource Identifiers (URIs)—a generalisation of an HTTP URL to identify things on the Web, (b) resource manipulation through representations—which implies that both the client and server can choose the representation of the resource such as JSON-LD, (c) all messages are self-descriptive—they contain all the information that the client needs for interpreting the message such as indicating that the "Content-type" is "text/html", and (d) Hypermedia As The Engine Of Application State (HATEOAS)—which refers to the fact that a response should include links to possible actions. These hypermedia controls are comparable with links to forms, which makes out-of-band documentation needless [65,67].

Second, the client-server model implies several clients that communicate with a server. The client performs an action on a Web resource—using the HTTP protocol—by sending a request to the server [65].

Finally, Stateless and Cacheable prohibit the server to store the state of the client application [65]. This implies that every client request contains the context. This has two advantages; (a) as the server does not need to store the state of the client applications, it can easily scale up, and (b) the requests can be cached, which lowers the load on the server (see Figure 1).

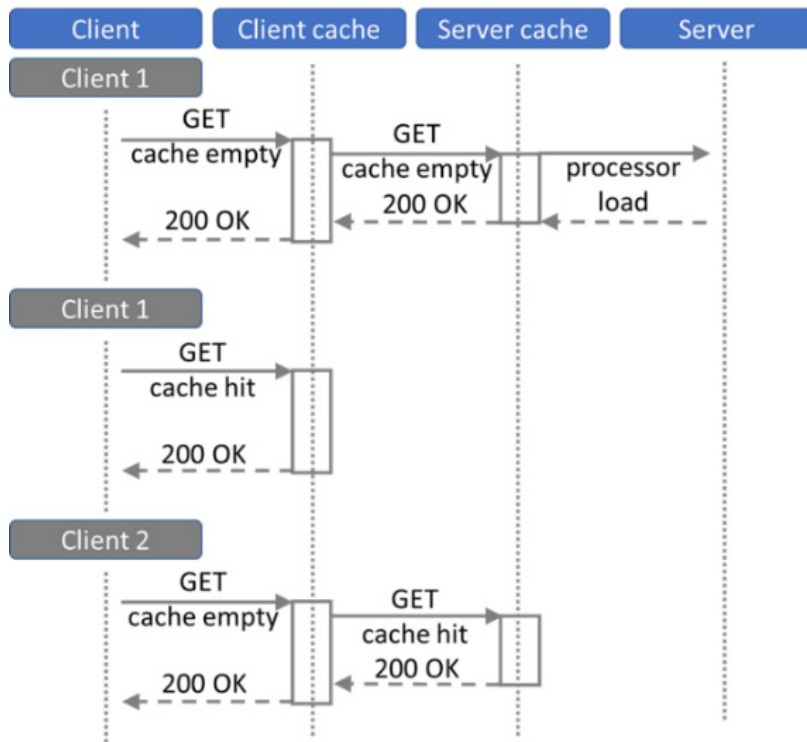

**Figure 1.** Different stages of caching: no caching, client caching and server caching [65,67].

*2.5. Balancing Efforts between Publisher and Consumers*

When servers encounter more complicated queries, they often respond with an error stating "query too complex", or "time out". Within the context of air quality data or railway infrastructure, this typically occurs in route planning use cases where end-users want to be routed only through areas where a specific property of air pollution is lower than a certain threshold or finding railway routes with specific parameter values (e.g., type of energy supply). Route planners should be able to perform such querying, without the data publisher ever having to think about this specific use case. A similar case, illustrating a route planner that can evaluate any given query on the client side, without having to rely on server-side functionality other than when downloading the right Linked Data Fragments, was implemented by Colpaert [68]. The application builds upon the principles of the REST architectural style and is an implementation of Linked Data Fragments (LDF).

LDF is a conceptual framework that provides a uniform view over Linked Data interfaces [69], including SPARQL endpoints, Linked Data documents and data dumps. All Linked Data interfaces publish specific fragments of a dataset that follow a certain *selector* pattern. This *selector* may be very specific, such as with a complex SPARQL query (materialized via a SPARQL endpoint), or very generic, such as in a single file data dump containing all the available triples/quads. Furthermore, in-between solutions exist, where such a data dump is fragmented, for example, based on geospatial characteristics [63]. A client can still answer individual queries by downloading the right subset of the knowledge graph. For a client to understand which fragments would be useful for answering a specific query, a server must document its fragmentation structure through hypermedia controls [69]. However, shifting query processing responsibility towards the client increases its complexity and may impact the query solving performance of certain types of queries due to additional data fetching tasks. This constitutes a trade-off that is also captured by the Linked Data Fragments axis (data dumps to SPARQL endpoints), which was first introduced for Triple Pattern Fragments, providing a low processing cost interface for answering Basic Graph Patterns at the expense of longer query response times.

To balance the effort between the data publisher and consumer of Air Quality Data, we limit the interface by applying a temporal and spatial fragmentation. This concept allows data to be published and consumed by moving intelligence from the server to the client, trying to create a better balance between the costs on the server and the client side [70]. When querying a dataset, an iterator allows the data container to be traversed, which is typically arranged as a tree or pipeline that divides the data stream into smaller parts that can be processed in parallel [71]. As we focus on self-describing and machine-readable data, we build upon the principles of Linked Data. Querying Linked Data is mostly associated with SPARQL Protocol and RDF Query Language, a semantic query language able to retrieve and manipulate the datasets. The approach of a dynamic iterator-based pipeline applied to process SPARQL queries has been researched [72]. SPARQL endpoints implement a protocol on top of HTTP—contrary to regular HTTP servers, there are many ways to express the same request that cache hits are likely to be very low—and, therefore, common HTTP caching cannot be used, which has a negative impact on the scalability [73,74].

The Linked Data Fragments approach leverages on HTTP caching and is, therefore, scalable. As time and space play a central role in air quality data, these are essential linking dimensions for LDF [25]. Examples of iterators are hydra:previous and hydra:next, which allow the client to iterate over the air quality time series, retrieving the different LDF samples at a particular timestamp or the average during a specified time interval. These hypermedia controls are defined in the Hydra Core Vocabulary [75]. These iterators were applied to time and space dimensions by Colpaert, who extended (https://open planner.team/specs/2018-11-routable-tiles.html, accessed on 26 July 2021) the Hydra ontology to describe a tile server that supports osm:Way, osm:Relation and osm:Node [68]. If an osm:Way has an overlap with a tile, links to bordering tiles will be added to the hydra:Collection (https://treecg.github.io/specification/, accessed on 26 July 2021).

## 3. Use Case Scenario

In this chapter, we will delineate and describe two use cases of publishing Linked Open Data time series. The first use case covers air quality data series captured by the delivery vans of a postal operator. The second use case explores an LDF-based architecture design for railway infrastructure data.

### 3.1. Air Quality Sensor Data Time Series

3.1.1. Overview

In order to delineate a clear use case, we conducted various semi-structured interviews with decision makers and experts. Interviews, as a methodological approach, can be structured, semi-structured or open ended, with the first usually employed within survey research and the latter in more explorative stages of research. Semi-structured interviews allow for more flexibility in which topic lists do not need to be followed rigorously and can be modified depending on the expertise or the issues raised during the conversation. According to Pfadenhauer, a semi-structured expert interview "lends itself as a data generating instrument in those cases in which the research focuses on the exclusive knowledge assets of experts in the context of their (ultimate) responsibility for problem solutions." [76]. We talked with experts at (a) the Flanders Environment Agency (https://www.vmm.be/, accessed on 26 July 2021) (VMM), an agency of the Flemish government working towards a better environment in Flanders, (b) the Agency for Facility Operations (https://overheid.vlaanderen.be/facilitairbedrijf, accessed on 26 July 2021) that is responsible for the Digital Archive Flanders, (c) Digital Flanders Agency (https://www.vlaanderen.be/digitaal-vlaander, accessed on 26 July 2021) that is responsible for digitisation and (d) the international innovation hub imec City of Things (https://www.imeccityofthings.be/en, accessed on 26 July 2021) that advances the state-of-the-art of smart city technology.

A use case was developed in which eighteen delivery vans of Belgium's leading postal operator were equipped with sensors to measure air quality on behalf of the University of Antwerp and imec (see Figure 2) [77].

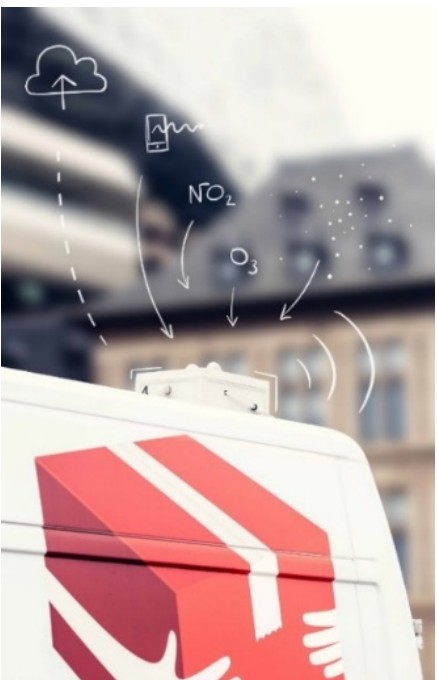

**Figure 2.** bpost van equipped with an air quality sensor (by imec City of Things).

Based on the gathered data, it is possible to suggest a healthier route with lower exposure to air pollutants to citizens. To realise this use case, multiple sensor data sources need to be queried. Therefore, we will also research a caching strategy that applies to Linked Data. The focus points are the northbound interfaces (NBI) for air quality analysis, which do not require real-time data streams [78–80].

The purpose of this use case was to evaluate the caching strategy. Evaluating techniques to calibrate the data based on information such as the sensor noise level or location is not in the scope of this use case.

As SOSA and SSN are, respectively, W3C recommendation and OGC implementation standards, and available as Linked Data, they are excellent candidates to facilitate IOP for air quality sensor data. Therefore, we have implemented SSN/SOSA for the use case scenario. However, we expect that, with the support of the European Commissions and communities including the International Data Spaces Association (https://www.internationald ataspaces.org/, accessed on 26 July 2021) and TM Forum (https://www.tmforum.org/pres s-and-news/fiware-foundation-tm-forum-launch-front-runner-smart-cities-program/, accessed on 26 July 2021), that NGSI-LD, and their "smart data models", could become a sustainable and interoperable standard for a wide variety of thematic domains.

### 3.1.2. Realising the Air Quality Use Case via a Linked Data Fragments Approach

In this section, we elaborate on how the use case scenario can be realised using an LDF approach. We developed three tracks (see Figure 3) that implement this use case, titled "the absolute sensor values in a time interval" (scenario 1), "the average sensor values per sensor" (scenario 2) and "the average sensor values within a bounding box" (scenario 3), which consume measurements $v_i$ at a certain timestamp $t_i$ from sensor nodes at a specific location $l_{sj}$.

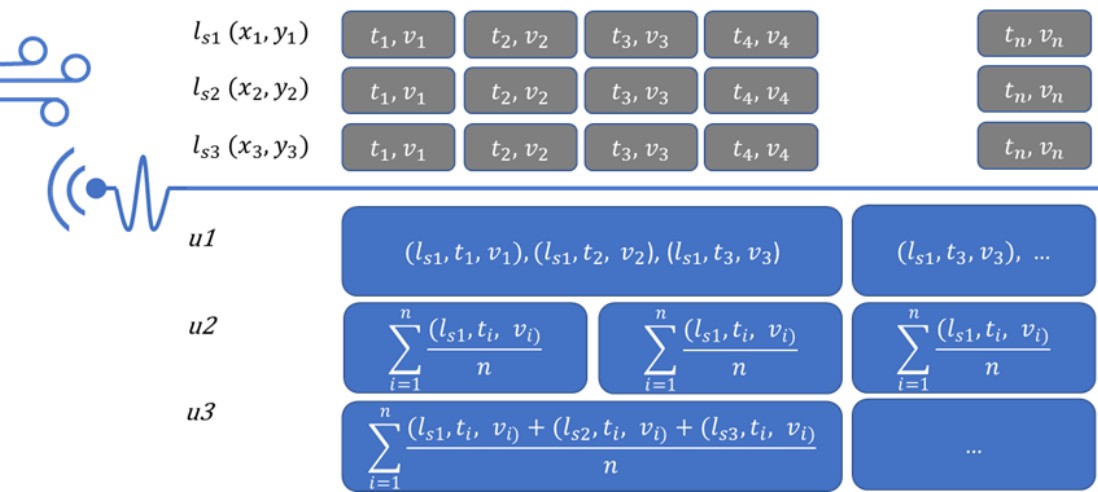

**Figure 3.** The scenarios (u1) "the absolute sensor values in a time interval", (u2) "the average sensor values per sensor" and (u3) "the average sensor values within a bounding box", which consume measurements $v_i$ at a certain timestamp $t_i$ from sensor nodes at a specific location $l_{sj}$.

The first track in the use case (scenario 1) is *titled* "the absolute sensor values in a time interval" and gives "the client" the option to request absolute sensor values in a time interval. The *primary actor* is the client of the Linked Time Series Server. The *main success scenario* consists of four steps. First, the client determines the time interval in which it wants to receive sensor values (1.1). This time interval can, for example, be an hour, day, month or year. Second, the client sends its request (1.2) with the following template: "http://example.org/data/\{z\}/\{x\}/\{y\}?page=\{timestamp\}" and the query parameter "page" containing the highest timestamp of sensor observations the client wants to retrieve. Third, the server responds with a fragment of the sensor observations

dataset based on the server's configured time fragmentation, for example, the observations must be fragmented per day, and the geographical tiling approach with zoom level "z", longitude tile "x" and latitude tile "y" (1.3). A fourth, optional step is needed when the client requires more data (1.4). In that case, the client needs to follow hypermedia links towards previous fragments. As the fragment from step 1.3 is provided with a hydra:previous attribute with a link to the previous fragment, the client can request this by simply following the link, which brings us to step 1.2. We identified two *extension scenarios*. Firstly, the client requests the most recent update. The client will receive the most recent fragment (1.1.a). Secondly, the server responds with an error message because no sensor values were found in the requested time interval; the system returns to step 1.1 (1.3.a).

The second track in the use case (scenario 2) has the *title*, "the average sensor values per sensor" and retrieves the average values of the air quality in a time interval. The *primary actor* is the client of the Linked Time Series Server. The *main success scenario* consists of three steps. First, the client determines the time interval from which he wants to receive average sensor values (2.1). This time interval can be an hour, day, month or year. Second, the client sends their request by using query parameters with the page timestamp, aggregation method and aggregation period. (2.2). Third, the server responds with the fragment corresponding with these parameters (2.3). We identified one *extension scenario*. 2.3.a The server responds with an error message because no fragment was found for the requested time interval, the system returns to step 2.1 (2.3.a). The third track in the use case (scenario 3) has the *title*, "the average sensor values within a bounding box". The client has the option to request a time series based on the average sensor values within an area defined by two longitudes and two latitudes. The *primary actor* is the client of the Linked Time Series Server. The *main success scenario* consists of four steps. First, the client determines from which bounding box he wants to request a time series (3.1). Second, the client sends their request (3.2). Third, the server responds with the desired fragment (3.3). Fourth, an optional step, the fragment from step three contains a hydra:previous attribute with a link to the previous fragment, the system returns to step 3.3. (3.4). Fifth, an optional step, the fragment from step three contains a hydra collection with a link to the neighbouring fragments, the system returns to step 3.3. (3.5). We identified no *extension scenario* for this scenario.

### 3.2. Railway Infrastructure Data through a Linked Data Fragments Approach

In order to assess the cross-domain applicability of our open data publishing approach, we explore the use of an LDF-based architecture design over railway infrastructure data. We focus on a common use case for this type of data, as is the calculation of routes over the railway network. We use the data managed by the ERA, which is given as a Knowledge Graph modelled on the ERA vocabulary (http://era.ilabt.imec.be/era-vocabulary/index-en.html, accessed on 26 July 2021).

The LDF approach is materialized on these data by creating highly cacheable geospatial data fragments of the railway infrastructure, in a similar fashion [58] to the road networks. However, in this case the geospatial tiles are produced on the fly via predefined SPARQL queries (see Figure 4), in contrast to the pre-processing tile generation carried out for the road networks. The geospatial tiling approach follows the Sippy Map (https://wiki.openstreetmap.org/wiki/Slippy_map_tilenames, accessed on 26 July 2021) specification, commonly used by vector tile-based map applications such as OSM. These tiles may be downloaded by client applications implementing a graph shortest-path algorithm (e.g., Dijkstra, A*), which will request the tiles starting from the input parameters of the route planning query, namely the origin and destination locations, and will perform a *follow-your-nose* approach to download more relevant tiles to answer a given query [81,82].

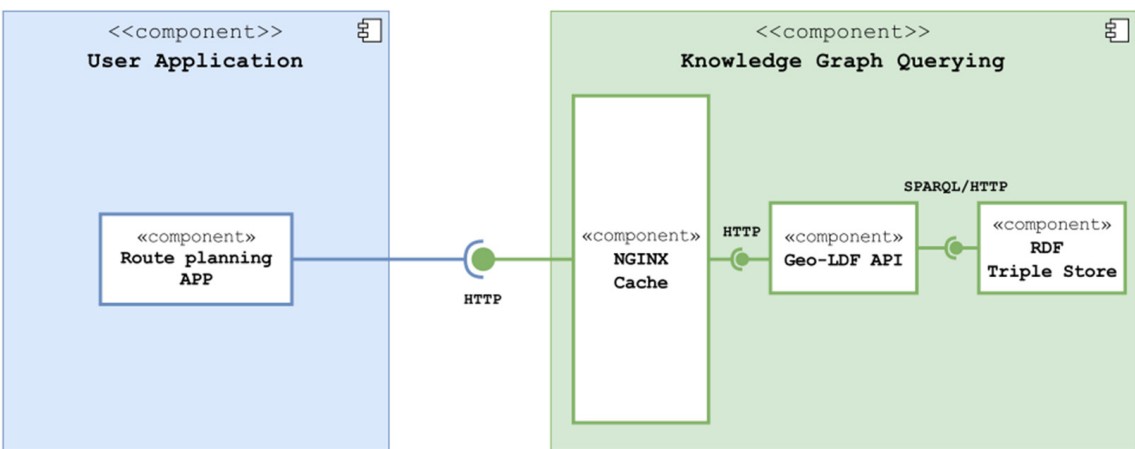

**Figure 4.** LDF-based architecture for railway infrastructure data.

Following the REST constraints, the data tiles are made available to client applications by means of an HTTP API. The API data responses contain self-descriptive metadata (hypermedia controls) that instructs clients on how they may find and request more relevant tiles to process their queries. Figure 4 shows a schematic diagram of this architecture. In Figure 5, we show an example of the hypermedia controls included in the API responses.

```
1 <http://era.ilabt.imec.be/sparql-tiles/abstraction/10/524/343>
2 tree:zoom "10"^^xsd:integer;
3 tree:longitudeTile "524"^^xsd:integer;
4 tree:latitudeTile "343"^^xsd:integer.
5 dct:isPartOf <http://era.ilabt.imec.be/sparql-tiles/abstraction>.
6
7 <http://era.ilabt.imec.be/sparql-tiles/abstraction>
8 a hydra:Collection.
9 dct:license <http://opendatacommons.org/licenses/odbl/1-0/>.
10 hydra:search [
11 a hydra:IriTemplate;
12 hydra:template "http://era.ilabt.imec.be/sparql-tiles/abstraction/{z}/{x}/{y}";
13 hydra:mapping
14 [
15 a hydra:IriTemplateMapping;
16 hydra:variable "x";
17 hydra:property tree:longitudeTile;
18 hydra:required "true"^^xsd:boolean
19 ],
20 [
21 a hydra:IriTemplateMapping;
22 hydra:variable "y";
23 hydra:property tree:latitudeTile;
24 hydra:required "true"^^xsd:boolean
25 ],
26 [
27 a hydra:IriTemplateMapping;
28 hydra:variable "z";
29 hydra:property tree:zoom;
30 hydra:required "true"^^xsd:boolean
31 ]
32 ]
```

**Figure 5.** Hypermedia controls and metadata for a railway infrastructure geospatial tile.

Similar to the air quality use case, caching plays a fundamental role in this architecture. Caching may be performed on the server side by means of the NGINX reverse proxy and on the client side, such as on the browser for Web applications. Due to client-side caching, one client will not need to request the same geospatial tile more than once, which can be re-used for processing independent queries. Server caching, on the other hand, allows a geospatial tile that has been already requested by one client to be served directly form the cache to other clients, freeing the RDF triple store of processing the same SPARQL query for every individual incoming request.

In Figure 5, we can see the metadata about the geospatial tile located at coordinates x:524, y:343, z:10, as specified by the Slippy Maps specification. We can see the description of these coordinates with the tree:longitudeTile, tree:latitudeTile and tree:zoom predicates (lines 2–4), which belong to the TREE hypermedia specification (https://treecg.github.io /specification/, accessed on 26 July 2021). The geospatial fragmentation is described by the template defining the X (longitude), Y (latitude) and Z (zoom) variables (lines 13–31). By fulfilling this template, a client application can request the railway topology data of specific regions, which can be used to perform route calculations.

To prove the feasibility of this approach, we implemented a client-side application that performs route calculations over the railway network, by traversing the different geospatially fragmented LDFs produced by the proposed architecture. Figure 6 depicts a screen capture of a route calculated using our client. The demo application is available online (http://era.ilabt.imec.be/compatibility-check-demo/, accessed on 26 July 2021). Evaluating the performance of this approach, considering, for example, the different shortest path algorithms and the different zoom levels, are among our next steps regarding this work.

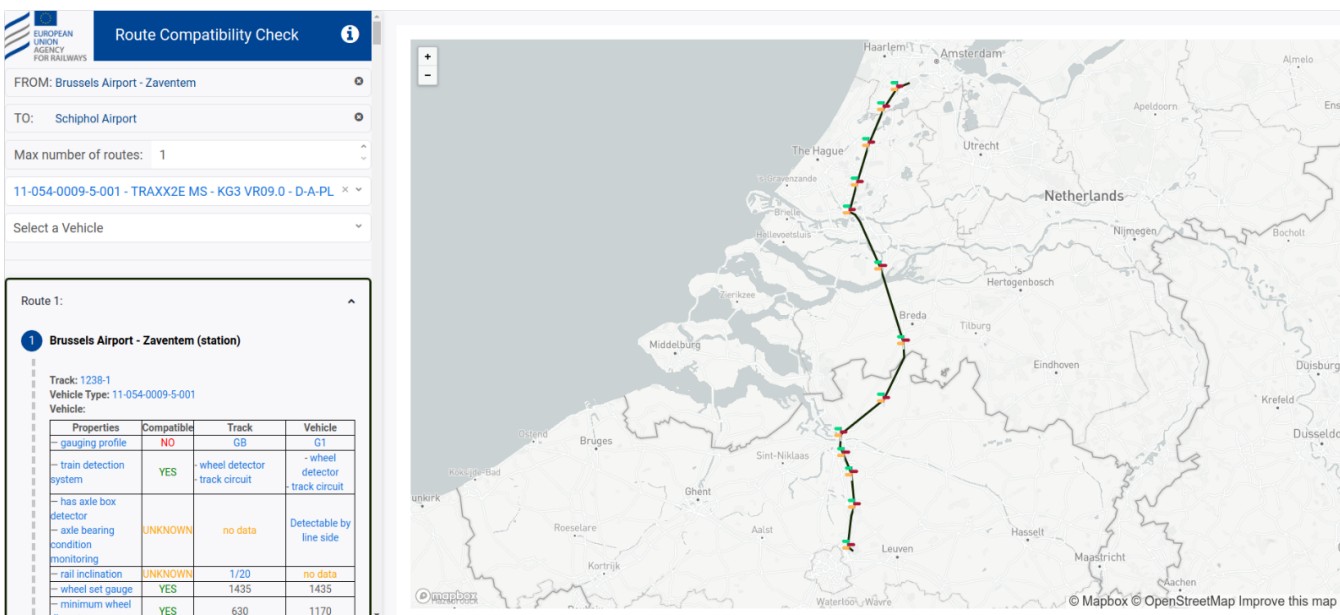

**Figure 6.** demo showing a route from Brussels airport to Amsterdam airport. © OpenStreetMap contributors (https: //www.openstreetmap.org/copyright, accessed on 26 July 2021).

## 4. Benchmark–Air Quality Data

### 4.1. Benchmark Characteristics and Approach

The goal of the benchmark is to test if the method of Linked Data Fragments on public endpoints for air quality sensor data will lower the cost for publishing and raise their availability due to a better caching strategy. We will compare the Linked Time Series (LTS) client/server setup to the FIWARE QuantumLeap (QL) API [83]. Therefore, we will monitor the parameters that characterise a Web API. According to Vander Sande [67], Web

APIs are characterised by (a) *the query response time, which refers to the rate at which tasks can be completed,* such as the maximum number of requests a server can handle in a time interval, (b) *the cost*, which indicates the amount of resources a single request consumes, such as the load on the CPU and memory of both the client and server, (c) *the cache re-use*, which is the ratio of items that are requested more than once from the cache and (d) *the bandwidth*, which is the required size of the HTTP communication.

In Section 3.1.2, we outlined three tracks to implement the use case that can be applied objectively to both architectural approaches. We will evaluate closed- and open-ended time intervals, as this has a significant impact on the caching strategy. To create an unbiased benchmark, we use the same database for both architectures. We add an extra scenario where we request the most recent observation, which provides a baseline without caching. The third scenario (scenario, the average sensor values within a bounding box) can be reduced to scenario two (the average sensor values per sensor) for this benchmark.

Hence, this leads to the following four benchmark scenarios:

- the most recent observation (b1);
- the absolute sensor values in a time interval that has not yet ended (b2);
- the absolute sensor values in a time interval that has ended (b3);
- the average sensor values in a time interval that has ended (b4).

To compare the LTS client/server setup with the FIWARE QL API, we performed load testing on both Web APIs by using the emulab (https://www.emulab.net/, accessed on 26 July 2021), which is a testbed that can be used for large networking and cloud experiments. The testbed consists of 160 pc3000 (https://gitlab.flux.utah.edu/emulab/emulab-devel/-/wikis/Utah%20Cluster#pc3000s, accessed on 26 July 2021) PC nodes with the following specifications: Dell PowerEdge 2850s with a single 3GHz processor and 2GB of RAM.

A synthetic dataset of observations is generated for one sensor. Observations are generated over a time span of five months, with one observation every ten minutes. As a result of using one sensor, only one location is considered for all the observations. The LTS server publishes one fragment per day and uses only one tile (/14/8392/5467) to geographically fragment the observations.

### 4.2. Testbed

As FIWARE is the preferred open-source platform by the European Commission, we will benchmark the Linked Time Series approach with the FIWARE QL API. First, we discuss the end-to-end architecture including the FIWARE QL API and the Linked Time Series, as illustrated in Figure 7 [84,85]. Next, we discuss the specific experimental setup, that uses the same back end for both architectures, to create an unbiased benchmark.

On the southbound, the IoT Agents (IoTa) facilitate the data stream from a sensor or a group of sensors to the context broker. These SBI interfaces typically use a native protocol. The Orion context broker is a building block of the FIWARE platform that decouples context producers and consumers. The broker facilitates updates, queries or subscription to changes on context information. The clients that subscribe are notified when specific conditions arise, such as a change in the air quality or location [86]. The context elements—in this experiment, air quality data—are stored in a document-based MongoDB database.

First, we evaluate the FIWARE QL API, which stores the data into a CrateDB time-series database. The data can be queried via a REST API that serves the space-temporal FIWARE-NGSI v2 (https://fiware.github.io/specifications/ngsiv2/stable/, accessed on 26 July 2021) data [83]. The second component in our experimental setup evaluates the Linked Time Series (LTS) Server, which is an implementation of LDF and is illustrated in Figure 7 (right).

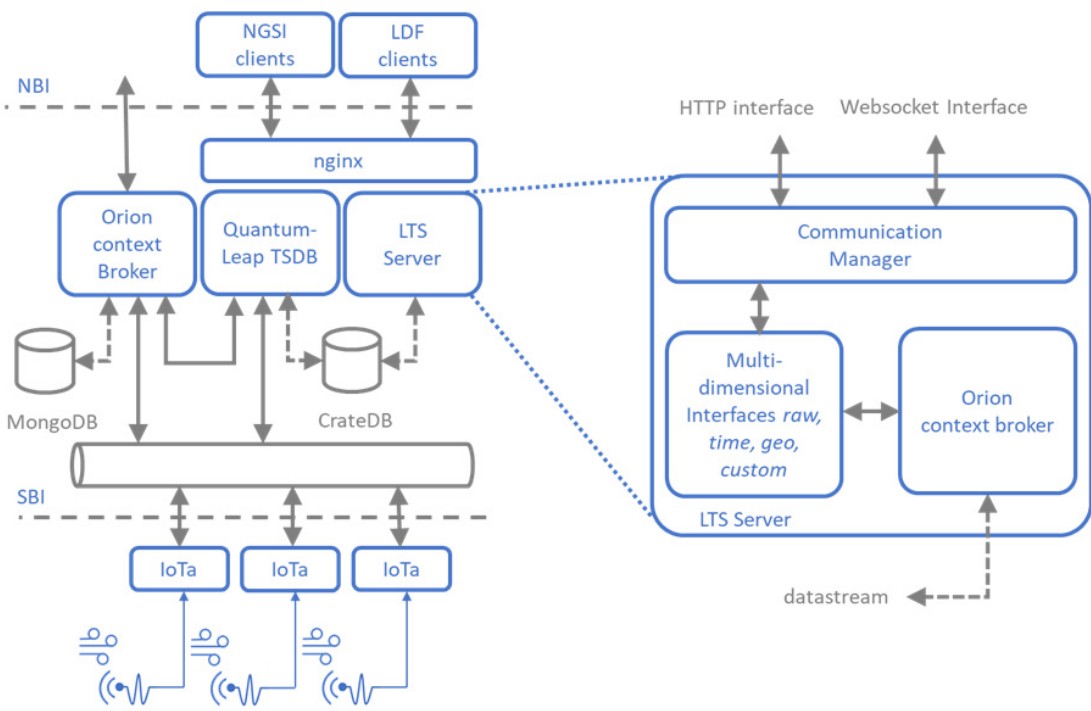

**Figure 7.** Overview of the end-to-end testbed with the FIWARE QuantumLeap API (**left**) and the Linked Time Series Server (**right**) [84,85].

We distinguish the following three main building blocks: (a) the Data Event manager that facilitates the stream updates, (b) the Multidimensional Interfaces that subscribe to specific events of the Data Event manager and calculate a predefined index and (3) the communications manager that facilitates the communication between the Multidimensional Interfaces and the clients [85]. In this experimental setup, both the QL API and LTS server use the CrateDB database, to create an unbiased benchmark. The LTS server provides an LDF interface for publishing time series and uses the method of Multidimensional Interfaces (MI) to fragment and index the data [87,88]. MI ensures the discoverability of the fragments by annotating them with hypermedia controls, which are formalised in an RDF vocabulary. The vocabulary (http://semweb.datasciencelab.be/ns/multidimensional-interface/, accessed on 26 July 2021) introduces the concepts Range Fragments and Range Gates. Taelman [87] defines a Range fragment as "an LDF that has an interval as a selector"—which is part of a predefined fragmentation strategy—and a Range Gate as "a Linked Data interface through which Range Fragments can be selected by interval" and, thus, exposes a collection of Range Fragments [87]. These Ranges were applied to the time and space dimensions of the air quality data and both fragment and index the absolute and average sensor values in a time interval.

Nginx (https://www.nginx.com, accessed on 26 July 2021) was added to serve as a Web cache (or HTTP cache), which stores copies of requests, for both the FIWARE QL API and LTS server.

We used Kubernetes—an open-source container-orchestration system—to package our testbed. One CPU, in Kubernetes (https://kubernetes.io/, accessed on 26 July 2021), is equivalent to one AWS vCPU, one GCP Core, one Azure vCore, one IBM vCPU or one Hyperthread on a bare-metal Intel processor with Hyperthreading. The results are expressed in mebibyte and millicpu. A mebibyte (MiB) is equivalent to 1048 576 bytes. Kubernetes defines a metric called Millicores that is used to measure CPU usage. It is a CPU core split into 1000 units. To ensure that the benchmark is reproducible, we have published the repository (https://github.com/brechtvdv/benchmark-quantumleap, accessed on 26 July 2021) with source code and configuration scripts. The repository provides the necessary scripts and background information to deploy and benchmark the FIWARE QL API with

the LTS Server API for timeseries on a Kubernetes cluster. We outline the main steps that are executed during the benchmark. First, we setup the Kubernetes cluster, a set of machines that run the containerised applications. Second, the scripts that deploy CrateDB, MongoDB, the Orion context Broker, QuantumLeap TSDB and the Nginx Web cache are executed. Third, the metrics server that harvests the CPU and memory consumption of the server and clients is deployed. Fourth, we setup the data streams by creating a subscription between the Orion context Broker and the QuantumLeap TSDB, which ensures automatic updates. Furthermore, a table is created in CrateDB, that stores the time series data. Fifth, data are ingested every second to Orion. Sixth, an HTTP client is provided and configured to use the QL and LDF API. Finally, the four scenarios are executed on both the QL and LDF API and monitored using the metrics server

### 4.3. Results

#### 4.3.1. The Most Recent Observations (b1)

This scenario benchmarks the request of the most recent observations (n = 100). The memory usage of the FIWARE QL API remains stable (Figure 8). Figure 9 shows that the CPU use of the FIWARE QL API and underlying database, with a load of ten clients, is a factor of four higher than the LTS Server API. At a load of four hundred clients that send a HTTP request every two seconds, we notice that the query response time of the FIWARE QL API increases to five seconds; at a higher load we obtain a timeout (Figure 10). Figure 10 shows an overview of the query response time (latency) clients perceive when using the FIWARE QL API. This contrasts with the LTS Server API that—despite a rising query response time—still stands with a load of 1300 clients (Figure 11). In both cases, caching cannot be applied, thus every request is passed on to the back end and results in one database request. In the case of the LTS Server API, a client is configured to retrieve all observations from the current day. The data are fragmented per day, meaning only one request will be sent. Based on the number of observations N that is returned, the client of the FIWARE QL API is configured to retrieve the equivalent last N observations by sending an HTTP GET request with path "/v2/entities/urn:ngsi-ld:Sensor:123?lastN=N". The bandwidth per request of the FIWARE QL API is 4.5KB, compared to 17.5KB for the LTS Server API. The load on the LTS Server API clients is 10 millicpu, compared to 2 millicpu in the case of the clients of the FIWARE QL API.

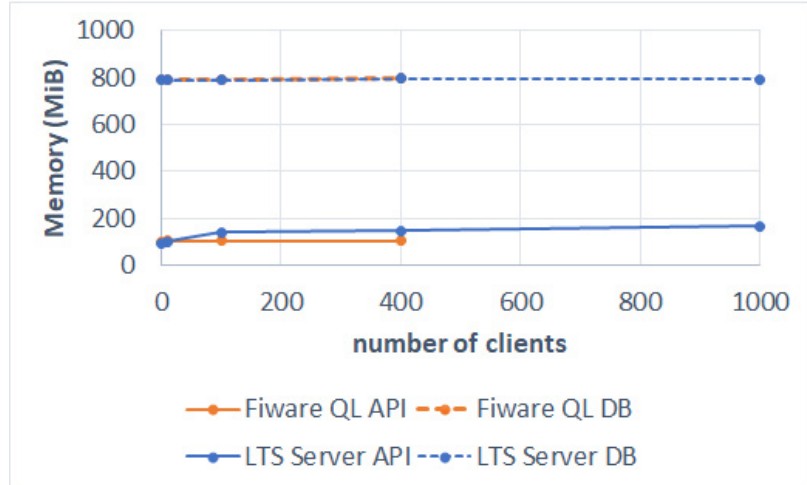

**Figure 8.** Overview of the benchmark with memory cost needed to publish the most recent observations.

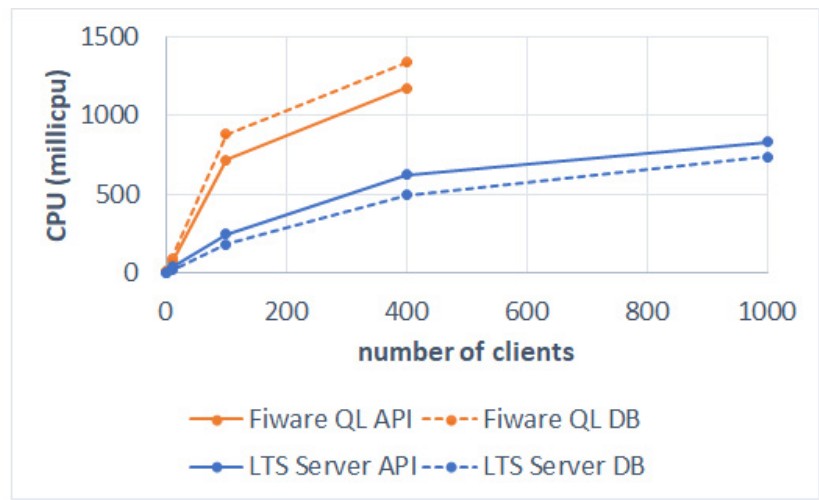

**Figure 9.** Overview of the benchmark with CPU cost needed to publish the most recent observations.

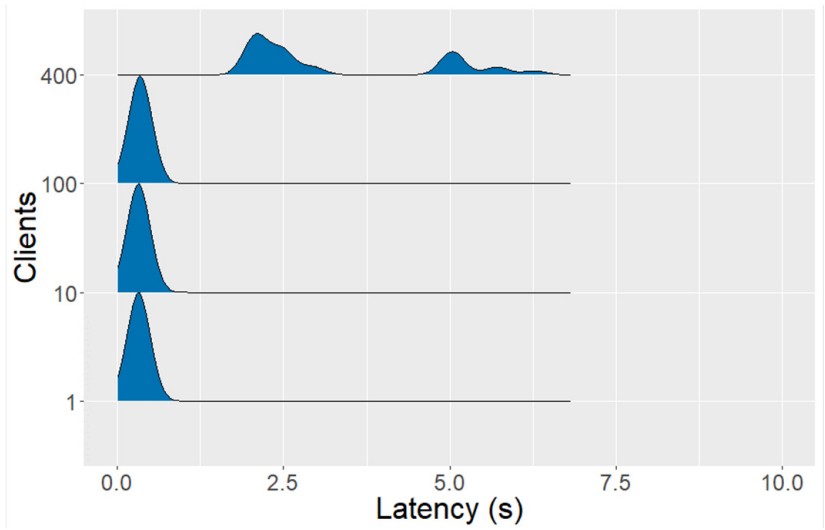

**Figure 10.** Overview of the benchmark with the query response time (latency) of FIWARE Quantum-Leap API, when publishing the most recent observations.

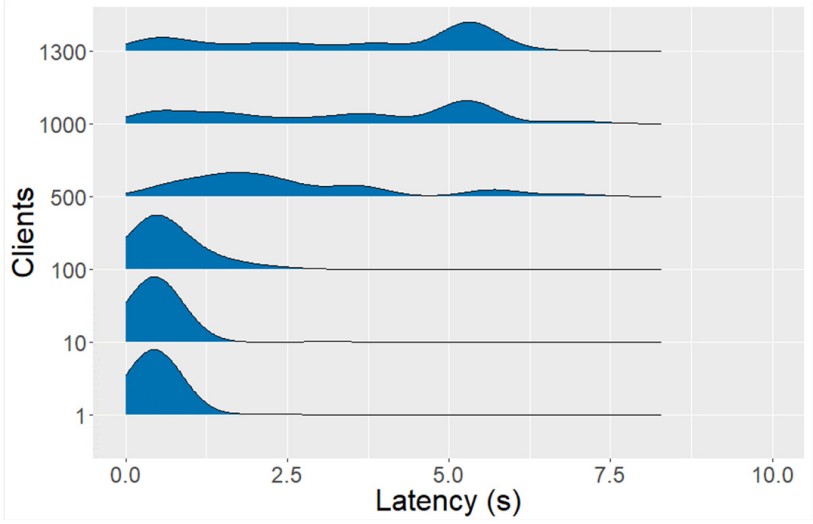

**Figure 11.** Overview of the benchmark with the query response time (latency) of Linked Time Series Server, when publishing the most recent observations.

### 4.3.2. The Absolute Sensor Values in a Time Interval That Has Not Yet Ended (b2)

This scenario benchmarks the absolute sensor values in a time interval (month) that has not yet ended. The memory usage of the FIWARE QL API remained stable (Figure 12). The CPU cost of the FIWARE QL API increased by a factor of twenty when scaling up to ten clients and double when scaling up from ten to one hundred clients (Figure 13). At a load of ten clients, the query response time of the FIWARE QL API remained below two seconds. From a hundred clients, the query response time rose above ten seconds. The query response time at a load of four hundred clients reached up to twenty seconds before a timeout from the FIWARE QL API was received (Figure 14). Figure 14 shows an overview of the benchmark with the query response time (latency) of FIWARE QL API and Figure 15 of the Linked Time Series Server. The CPU cost of the FIWARE QL API database increased by a factor of fifteen when scaling up to ten clients and doubled when scaling up from ten to one hundred clients (Figure 13). The CPU cost of the FIWARE QL API client was almost neglectable. Since there is no client-side cache re-use, the bandwidth of the FIWARE QL API was 189.6KB, for an interval of one month.

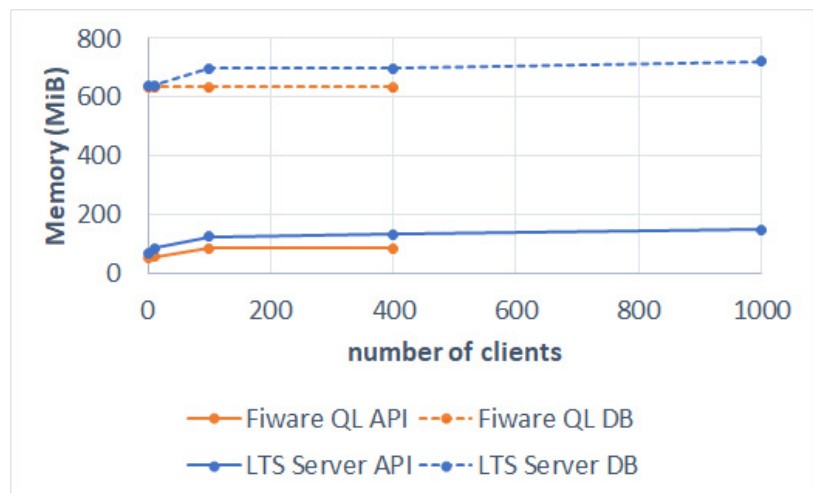

**Figure 12.** Overview of the benchmark with the memory cost to publish the absolute sensor values in a time interval that has not yet ended (b2).

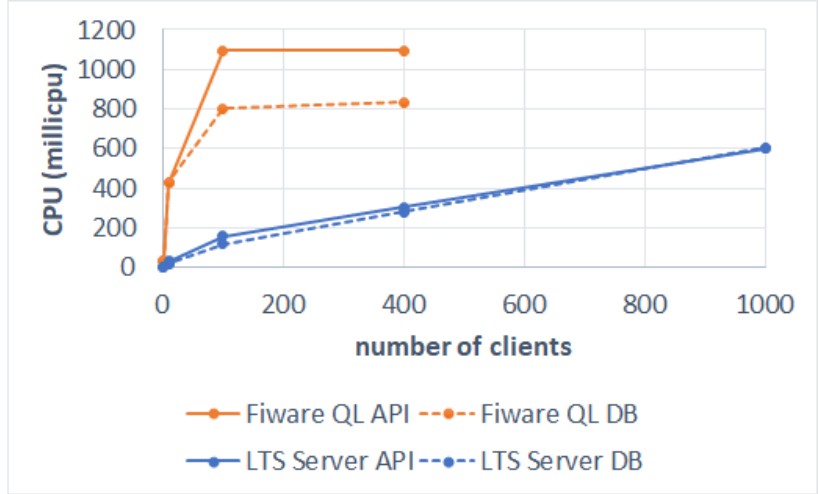

**Figure 13.** Overview of the benchmark with the CPU cost to publish the absolute sensor values in a time interval that has not yet ended (b2).

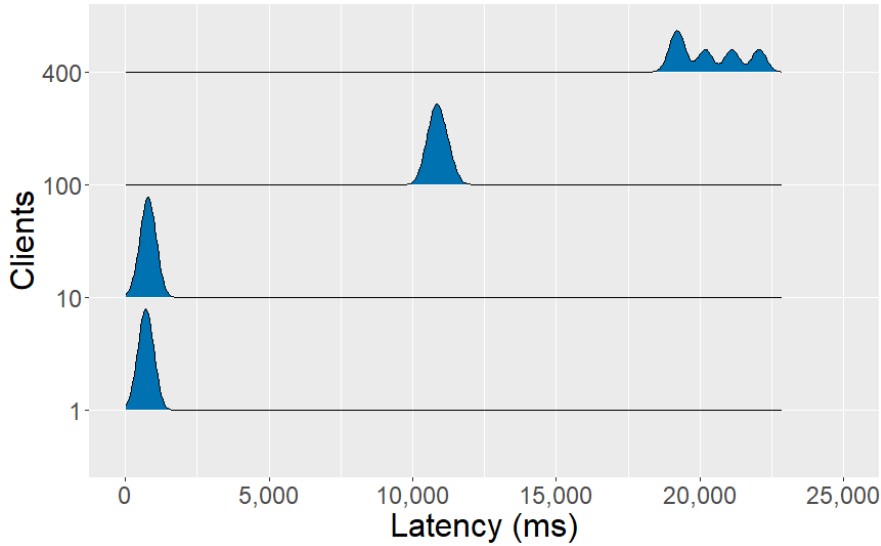

**Figure 14.** Overview of the benchmark with the query response time (latency) of FIWARE Quantum-Leap API, publishing the absolute sensor values in a time interval that has not yet ended (b2).

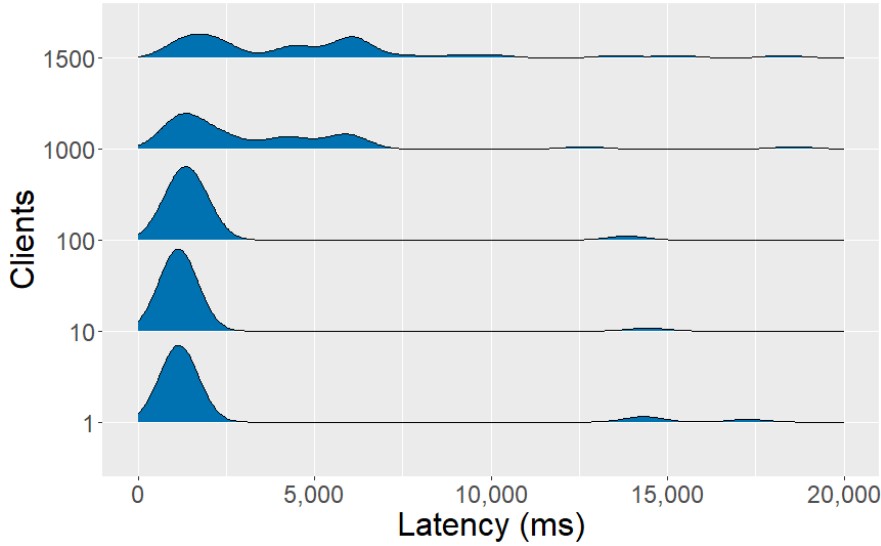

**Figure 15.** Overview of the benchmark with the query response time (latency) of Linked Time Series Server, publishing the absolute sensor values in a time interval that has not yet ended (b2).

The CPU cost of the LTS Server API, increased linearly with the number of clients (Figure 13). From the second request, there was cache re-use of 30/31, as the first request could not be cached. The CPU load on a thousand clients was lower than a load on ten clients in the case of the FIWARE QL API. The latter needed to run for every client's request for a database query ranging all 31 days, while the former had the benefit of 30 HTTP cache hits with NGINX. The load on the clients was 387 millicpu, compared to 2 millicpu in the case of the clients of the FIWARE QL API. The bandwidth at the LTS Server level was 542.2 KB per client, with a cold client-side cache. From the second query onwards, the bandwidth dropped to 17.5KB due to client caching.

### 4.3.3. The Absolute Sensor Values in a Time Interval That Has Ended (b3)

This scenario benchmarks absolute sensor values in a time interval (month) that has ended. The results of the FIWARE QL API are comparable with those of the absolute sensor values in a time interval that has not ended yet (Figures 16 and 17). From a hundred clients, the query response time of the FIWARE QL API rose above ten seconds. The query

response time at a load of four hundred clients rose to twenty seconds before receiving a timeout (Figure 18). Figure 18 shows an overview of the benchmark with the query response time (latency) of the FIWARE QL API and Figure 19 of the Linked Time Series Server. As the time interval had ended, all 31 responses of the LTS Server API were fully cacheable. This allows up to 1000 clients without an additional CPU cost or increases in memory (Figure 17). The load on the clients was 401 millicpu, compared to 5 millicpu in the case of the clients of the FIWARE QL API. The bandwidth at the LTS Server level was 524.9 KB per client. From the second request, this dropped to 17.5 KB.

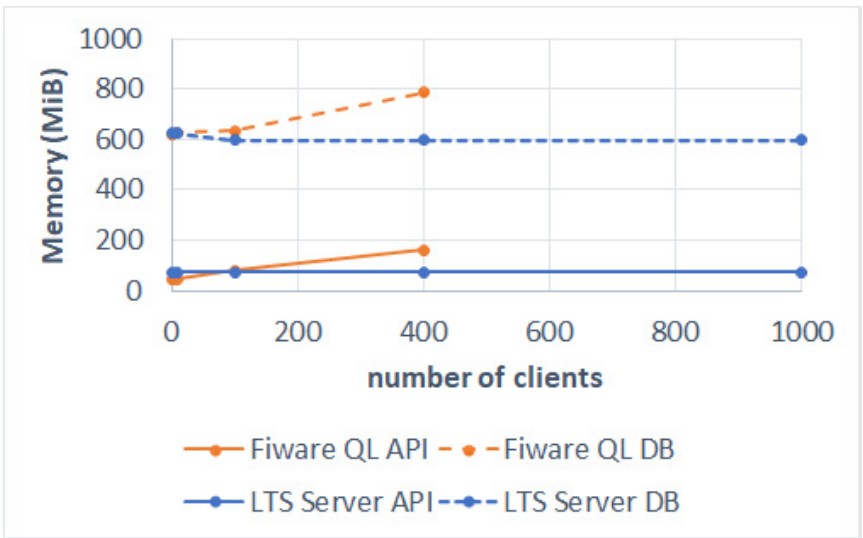

**Figure 16.** Overview of the benchmark with the memory cost to publish the absolute sensor values in a time interval that has ended (b3).

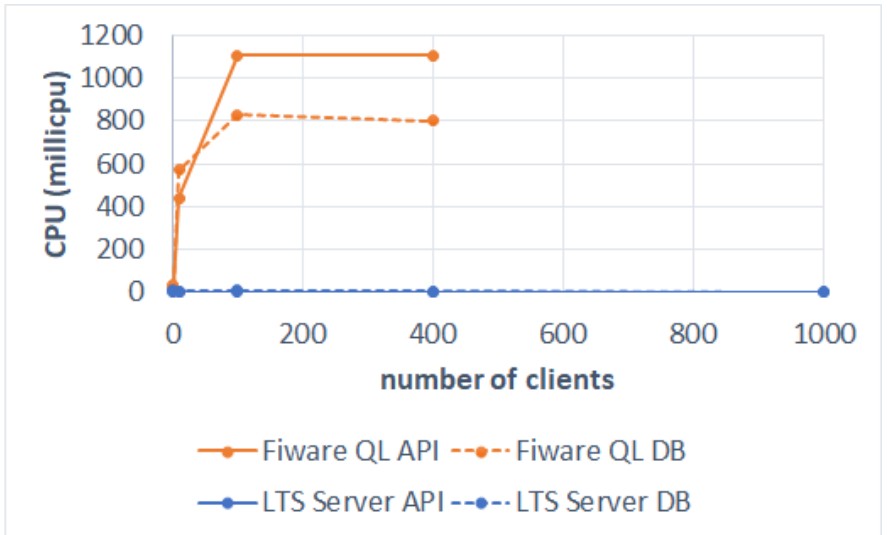

**Figure 17.** Overview of the benchmark with the CPU cost to publish the absolute sensor values in a time interval that has ended (b3).

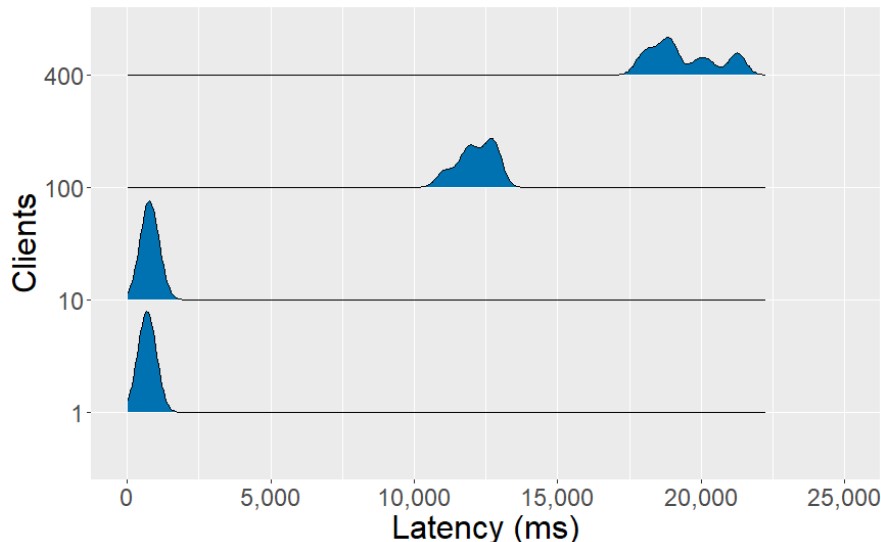

**Figure 18.** Overview of the benchmark with the query response time (latency) of FIWARE Quantum-Leap API, publishing the absolute sensor values in a time interval that has ended (b3).

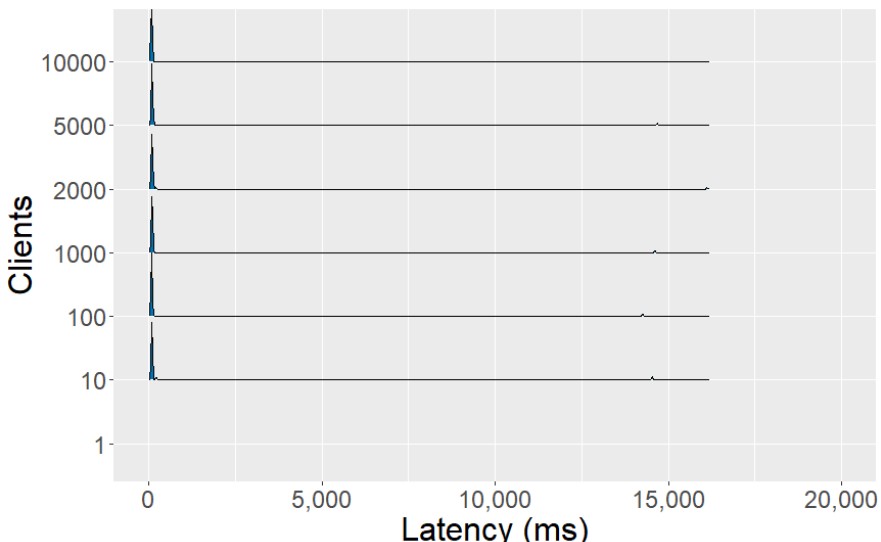

**Figure 19.** Overview of the benchmark with the query response time (latency) of Linked Time Series Server, publishing the absolute sensor values in a time interval that has ended (b3).

4.3.4. The Average Sensor Values in a Time Interval (Hour) That has Ended (b4)

This scenario benchmarks the average sensor values in a time interval (hour) that has ended. As the time interval had ended, the responses of the LTS Server API were fully cacheable. This allowed up to ten thousand clients with only a slight increase in the CPU cost and without any increase in memory (Figures 20 and 21).

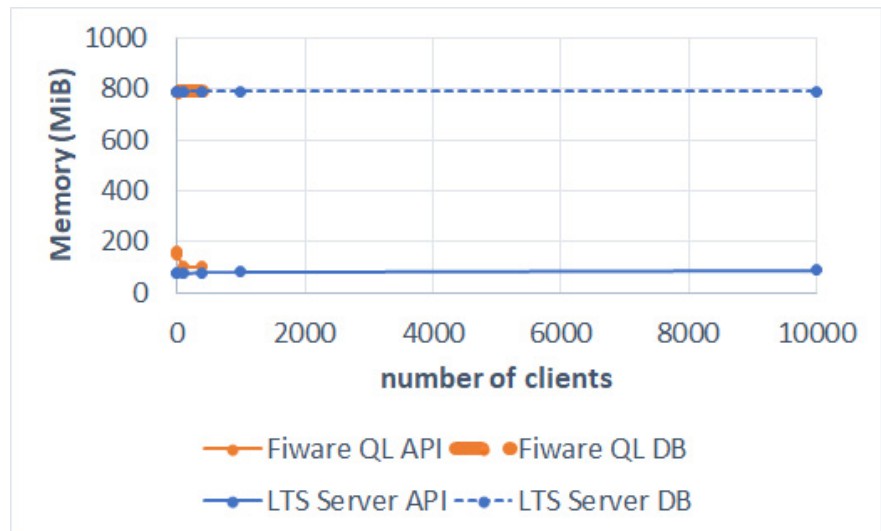

**Figure 20.** Overview of the benchmark with the memory cost to publish the average sensor values—stretched to 10,000 clients—in a time interval (hour) that has ended (b4).

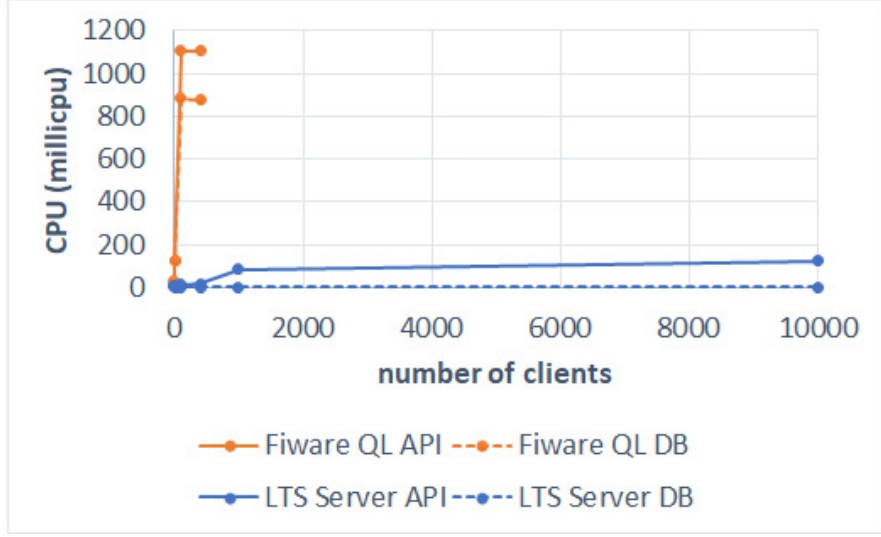

**Figure 21.** Overview of the benchmark with CPU cost to publish the average sensor values—stretched to 10,000 clients—in a time interval (hour) that has ended (b4).

The results of the FIWARE QL API are comparable with those of the absolute sensor values in a time interval that has not ended yet (Figures 22 and 23). From four hundred clients, the query response time of the FIWARE QL API increased to above ten seconds. The query response time at a load of four hundred clients increased to twenty seconds before receiving a timeout (Figure 24). Figure 24 shows an overview of the benchmark with the query response time (latency) of FIWARE QL API and Figure 25 of the Linked Time Series Server.

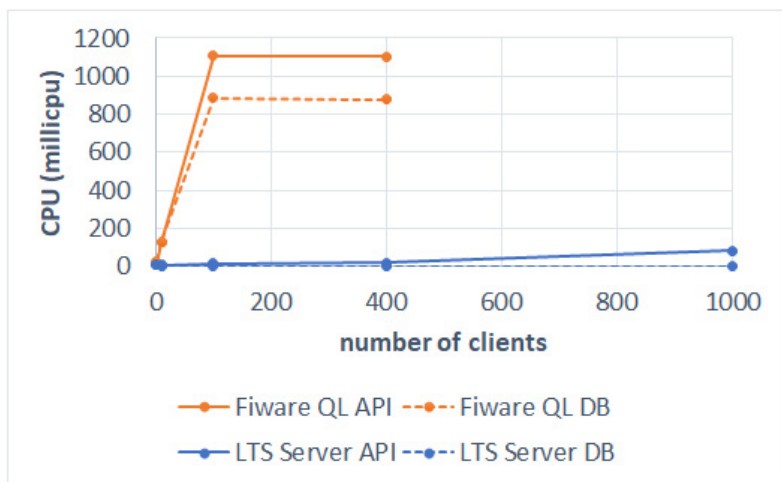

**Figure 22.** Overview of the benchmark with the CPU cost to publish the average sensor values in a time interval (hour) that has ended (b4).

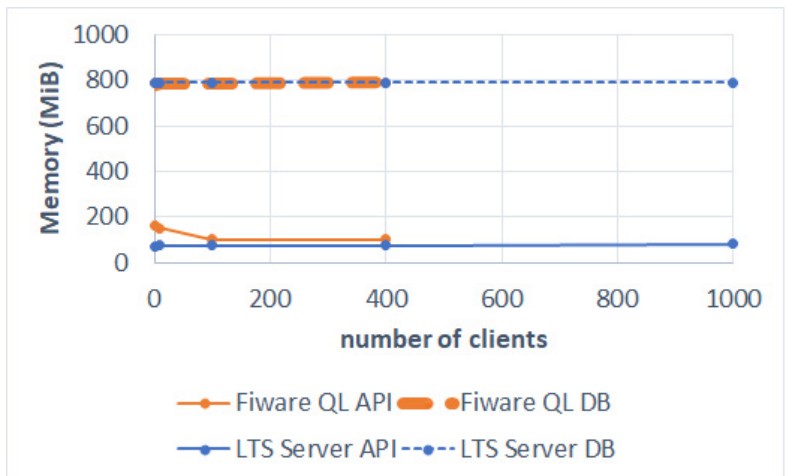

**Figure 23.** Overview of the benchmark with the memory cost to publish the average sensor values in a time interval (hour) that has ended (b4).

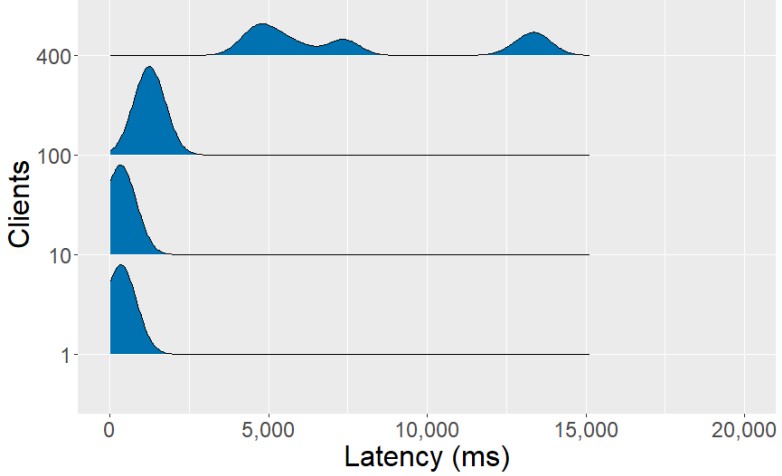

**Figure 24.** Overview of the benchmark with the query response time (latency) of FIWARE Quantum-Leap API, publishing the average sensor values in a time interval (hour) that has ended (b4).

The load on the clients of the LTS Server API is 367 millicpu, compared to 2 millicpu in the case of the clients of the FIWARE QL API. As the clients of the LTS Server API need to calculate the average, their load is significantly higher. The bandwidth at the server level is 524.9 KB per LTS Server API client—as the server responds with all the data from the last month—compared to 38.5 KB for the FIWARE QL API that only responds the pre-processed average sensor value to the client. From the second request, the bandwidth of the LTS Server drops to 17.5 KB.

## 5. Discussion

### 5.1. Air Quality Sensor Data Time Series

Government administrations as data providers make significant investments in collecting data, making it interoperable and publishing it for maximum re-use. Hence, our research question addressed how data providers can develop a sustainable method for publishing open sensor data, in specific sensor data time series on air quality. In order to do this, we set-up a benchmarking experiment. We benchmarked the QL API and a Linked Time Series API, which are running on the same database.

The first scenario, retrieving the most recent observations, shows that the CPU usage of both APIs increases linearly with the number of clients (Figure 9). However, the QL API increases more steeply. Furthermore, the LTS API can serve more than three times the number of clients before timing out (Figures 10 and 11). A possible explanation for this might be that QL API's code base uses more abstraction layers and, thus, can be less performant than the LTS API code base.

In the second scenario, where the absolute sensor values over one month, inclusive of the most recent observations, are fetched, the difference in CPU usage is even clearer (Figure 13). This result may be explained by the fact that the QL API retrieves one request with a time interval of one month, while the LTS API retrieves one request per day. This finding suggests that limiting querying capabilities to retrieving fragments is important for the sustainability of a sensor data API. This fragmentation approach also allows fragments with historic data to be cached on the client side lowering the number of requests and load on the server. As a result, the query response time of the LTS API is slightly higher for a few clients but remains acceptable for a higher number of clients (Figures 14 and 15).

The third scenario refines the second scenario by only retrieving historic data. The results on the CPU, memory usage and query response time of the QL API (Figures 16–18) indicate similar results as the previous scenario. The LTS API, conversely, benefits from the cacheability of all the fragments: its CPU cost becomes negligible (Figure 17) and the query response time on the client remains below 150ms, except for the first run with a cold cache (Figure 19).

Finally, the last scenario refines the third scenario by calculating average values per hour instead of returning the raw sensor values. We see an increase from 800 to 850 millicpu usage in the database with the QL API (Figure 22 versus Figure 17). This result may be explained by the fact that aggregation queries are more complex than select queries. One unanticipated finding was that the query response time of the QL API has improved significantly in this scenario: at a load of 100 clients, the query response time is still below 2.5 s (Figure 24). The results on the LTS API are similar to the previous scenario due to the ended interval allowing the fragments, which contain the aggregated values, to be fully cacheable (Figure 25).

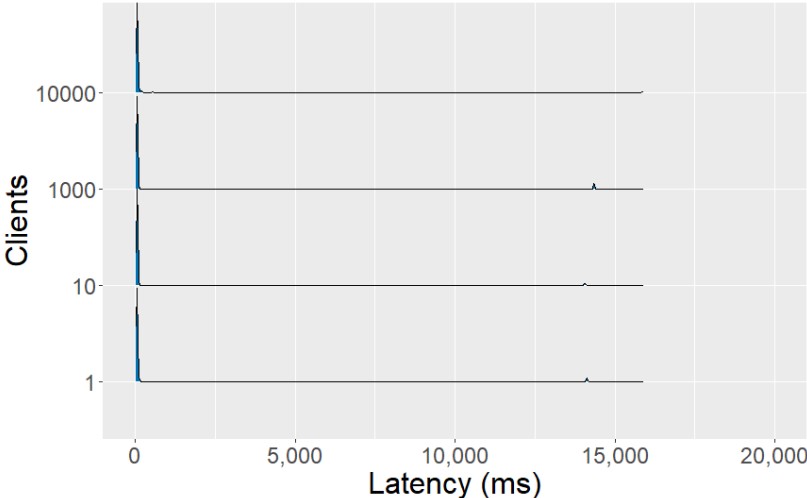

**Figure 25.** Overview of the benchmark with the query response time (latency) of Linked Time Series Server, publishing the average sensor values in a time interval (hour) that has ended (b4).

The benchmark showed that the Linked Time Series approach lowers the cost for publishing air quality data and raises the availability because of a better caching strategy. The results of the benchmark ascertain that—with an increasing number of clients—the LTS API has a lower CPU load than the FIWARE QL API. Additionally, as the load of the LTS is constant for historic data, the cost becomes predictable, which is crucial for public bodies that need to determine their expenses beforehand. Building on the strength of the World Wide Web—by using HTTP caching on the level of clients and servers—created not only a cost–benefit but also contributed to the stability of the air quality endpoints. The FIWARE QL API starts "sputtering" at a load of 400 clients (Figure 24), in contrast with the LDF interface that still obtains good results at 10,000 requests (Figure 25).

The bandwidth of the LDF endpoint is slightly higher, but the number of resources a single request consumes—such as the load on the CPU and memory—is significantly lower, due to the cache re-use. The cache re-use is the ratio of items that are requested more than once from the cache instead of consuming server resources. The literature is consistent with our findings from limiting the server interface that becomes more scalable due to Web caching [65,69]. The scenario that calculates the average sensor values in a time interval that has ended demonstrates that a part of the workload is shifted to the client but is still acceptable for the client. A non-measurable benefit is that the client can evaluate any given query on the client side, without having to rely on server-side functionality other than downloading the right fragments. Next, the Open World Assumption (OWA) becomes applicable: more data can always be downloaded in order to obtain a more precise answer [89,90].

We discussed that the challenges related to linked open sensor data time series are not limited to the volume and velocity of the time series but also to their variety interoperability (IOP) challenges, which are crucial when combining air quality data from different sources as well as linking them to other datasets such as traffic or weather data. We addressed the different IOP levels, namely the legal, organisational, technical and semantic level. Linked Data facilitates IOP on both a technical and a semantic level. Context information, such as temperature, humidity and spatial information, enriches air quality data by re-using existing machine-readable RDF vocabularies. The principles of Linked Data make the data self-describing and machine-readable, which allows autonomous agents to reason on the sensor data. Linked Data builds upon the architecture of the World Wide Web and uses typed links between data entities from disparate sources, described using the Resource Description Framework.

Based on the above, we have three recommendations about archivability, indexing and interoperability for future research. First, it is crucial to ensure that time series are

still accessible and usable for future generations, as they are valuable for research (e.g., on environmental changes). Therefore, a strategy that outlines which subsets of the data should be preserved in order to reduce the storage cost of the data in a digital archive needs to be defined. Second, research should consider a more dynamic method to fragment and index different types of time series, considering the available budget of the publisher. Finally, on the level of interoperability, future research should explore how to bridge between the NGSI—that redefines a knowledge representation in its own—and the existing semantic assets.

Although our results are promising, there are limitations to our research. First, as the scenarios are limited, further validation of this method in a large variety of use cases and different types of sensor data time series is necessary to extrapolate our conclusions. Second, we should evaluate this method for real-time data streams.

During this research, we determined extra challenges related to cross-domain interoperability and architectural flexibility, which we discuss in the next paragraph.

### 5.2. Railway Infrastructure Data

Similar to public authorities publishing sensor-based data, ERA seeks to publish the European railway infrastructure data in an interoperable way to maximize re-use by different applications serving diverse use cases. To increase interoperability, the ERA follows the Linked Data principles and publishes a knowledge graph containing the different elements that conform to the railway infrastructure, while including semantic annotations.

We showed that publishing railway infrastructure data is feasible with an LDF-based approach, bringing the same cost-efficiency and scalability benefits as the ones measured by our benchmark on air quality sensor data. In this case, we applied the same principles of well-known vector tiling strategies for geospatial data, but we further enriched the tiles with semantic annotations based on hypermedia description vocabularies (hydra and TREE). Such annotations enable client applications to interpret the data interfaces and to traverse the underlying knowledge graph.

Moreover, our proposed data publishing approach enables route calculations to be performed on the client side, while aiming to increase the cacheability of the server requests and including semantic annotations describing the data interfaces via hypermedia controls [91]. This architectural design allows for greater flexibility in client application design at the cost of increased complexity for application implementation. Developers are able to implement and customize any kind of algorithm and business logic, tailored to their use cases. This is in contrast to the dedicated server-side solution systems that limit the application to the supported capabilities and features.

### 6. Conclusions

In this article, we presented insights on the implementation of a sustainable method for publishing open data, in a specific sensor data time series on air quality and railway infrastructure data.

Our study demonstrates how Linked Data can support *interoperability* at the technical and semantical level for an air quality time series. Linked Data principles not only provide interoperability towards external stakeholders but also foster a more sustainable and cost-effective architecture. We have shown that exposing Linked Data Fragments for a sensor data time series lowers the publishers' server cost compared with the FIWARE QuantumLeap API. By monitoring the query response time of the client, we showed that the Linked Time Series (LTS) interface can support more clients (10,000 versus 400 clients with historic data), while having a lower latency. Especially for the time series queries where the time interval has ended, HTTP caching lowers the server CPU cost significantly. As such, the LTS interface can serve as a valuable extension of the FIWARE stack to provide *scalability* when publishing data on the Web.

The pilot on railway infrastructure data shows that this architectural approach—which shifts the data integration to the client—facilitates *flexibility*. Clients are able to perform any further processing on the data to support specific use cases. In our case, the client implements a shortest path finding algorithm, which can be tailored and adjusted for more specific needs unlike traditional server-side APIs, which limit clients to their supported features and algorithms. Moreover, context can be added at the client level without the need for rewiring the server, which often affects the entire ecosystem. However, such architectural design imposes a heavier burden on the clients, which may be reflected as poorer performance on query solving tasks. Optimized caching strategies are required to improve the overall performance of the addressed use cases.

The main difference between the two different explored use cases, lies in the nature of the data fragmentations that were applied to each case. On the one hand, for the air quality data, a time-based fragmentation was applied, which was aligned with the nature of the data and with the query requirements that needed to be supported. On the other hand, for the railway infrastructure data, we applied a geospatial fragmentation that again suited the query requirements brought forth by the use case we wanted to support. Besides the difference in fragmentation criteria, both approaches are built following the same architectural principles, where caching plays a fundamental role in reducing the operational costs of data interfaces. We, thus, showed how this approach can be applied to independent domains and over different types of data and demonstrated how data providers can develop a sustainable method for publishing open data.

According to President von der Leyen, common data spaces are an enabler for innovation and new jobs [8]. Interoperability within and between European common data spaces will be crucial to avoid data silos and increase the re-use of data [92,93]. We hope that insights from this article can speed-up the process of opening datasets by public and private organisations on a Web-scale and can catalyse the European Data Strategy. As such, our contributions, which build upon the principles of Linked Open Data, can be valuable for governments, organisations and researchers wanting to publish interoperable data on a Web-scale in a cost-efficient and flexible way.

**Author Contributions:** Conceptualization, R.B. and D.V.L.; methodology, E.V. and R.B.; software, B.V.d.V. and J.R.M.; validation, R.B. and B.V.d.V.; writing—original draft preparation, R.B., D.V.L. and J.R.M.; writing—review and editing, P.C., P.M. and M.V.C.; visualization, R.B. and J.R.M.; supervision, P.M., S.L. and E.M. All authors have read and agreed to the published version of the manuscript.

**Funding:** This research received no external funding.

**Institutional Review Board Statement:** Not applicable.

**Informed Consent Statement:** Not applicable.

**Acknowledgments:** The research activities presented in this article were funded by imec—IDLab Ghent University, Belgium and imec City of Things, Belgium. We also would like to thank Quincy Oeyen from the Agency for Facility Operations (responsible for the Digital Archive Flanders), Christophe Stroobants from the Flanders Environment Agency, Ziggy Vanlishout from the Digital Flanders Agency and Philippe Michiels from imec City of Things for their valuable insights. We also extend our gratitude to the team at ERA for facilitating access to their data. Finally, we would like to thank Esther De Loof and Arthur Macpherson for proofreading this manuscript.

**Conflicts of Interest:** The authors declare no conflict of interest.

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
