# Peer review of "A Sustainable Method for Publishing Interoperable Open Data on the Web"

_data, 2004_

Round 1

Reviewer 1 Report

30: these fast-growing megacities

urban areas do not automatically mean megacities.

93: In this article, we address the case of cost-efficient publishing of sensor data time series on air quality we explore the case of cost-efficient open data publishing of the European railway infrastructure Therefore, our main research question addresses how public authorities as data providers can develop a sustainable method for publishing open data, in concreto sensor data time series on air quality and railway infrastructure data.

Yet no mention of the OGC SensorThings API or any other standards for sharing time series data?

These barriers, referred to as legal IOP

How is a barrier an example of Interoperability

Table 1 is not referenced in the text.

235: The competing vocabularies which model the domain of air quality from slightly different viewpoints, including INSPIRE, NGSI-LD and SSN/SOSA are discussed (see Table 2).

Except they are not discussed, only an overview is given in Table 2.

290: As railway infrastructure data is a networked ecosystem where various infrastructure managers — such as Infrabel in Belgium — interoperability is crucial

Broken sentence.

367: 2.4. Balancing efforts between publisher and consumers

This section does not actually address this "balancing". When queries are made simpler for the server, they become more complex for the client, but this aspect is neglected.

380: It makes the observation that all Linked Data interfaces have in common that they publish specific fragments of a dataset, whether very specific, such as with SPARQL endpoints, or very generic, such as in a single data dump.

Unclear example. How is a data dump more generic than a SPARQL endpoint that has the same data?

390: To balance the effort between the data publisher and consumer for Air Quality Data, we limit the interface by applying a temporal and spatial fragmentation

It it not explained if or how this approach balances any effort.

398: SPARQL endpoints implement a protocol on top of HTTP and therefore common HTTP caching cannot be used.

It is quite possible to implement a protocol on top of HTTP and still make use of HTTP caching.

402: This approach leverages on HTTP caching and is therefore scalable.

Which approach? (unclear back-reference)
Why is the suggested approach better cacheable? This seems to be a key premise in the paper, but is not explained at all.
What is the API that is used? What requests are done? What responses are given?

423 & 430: Based on the gathered data, it is possible to suggest citizens a healthier route with lower exposure to air pollutants...

This paragraph is duplicated.

441: As SOSA and SSN are respectively a W3C recommendation and OGC implementation standards, and available as Linked Data, they are an excellent candidate to facilitate IOP for air quality sensor data. Therefore, we have implemented SSN/SOSA for the use case scenario. However, we expect that with the support of the European Commissions and communities including the International Data Spaces Association 15 and TM Forum 16 that NGSI-LD could become a sustainable and interoperable standard for a wide variety of thematic domains.

This comparison is odd. SOSA/SSN are data model standards, NGSI is mainly an API standard.

Examples of how SSN/SOSA is used are missing.

461: Second, the client sends its request

What does this request look like?
What is in a fragment?

461: Third, the server responds with a fragment

Does the server always respond with a single fragment? What if the required data spans multiple fragments? The used API is not described at all.

462: Third, ...

Steps 1.3 and 1.4 are not numbered

463: the client can request this by simply following the link

In what use-case would a client want the previous data? It already specified the time-frame of interest in 1.1

489: the server responds with the desired snippet

What is a "snippet"? The same as a Fragment? or something else?

490: contains a hydra: the previous attribute

There is a misplaced "the" in "hydra: the previous"

526: http://era.ilabt.imec.be/

You may want to check your server for security problems before publishing its URL in a publication. People will try the URL, and currently your GraphDB has "Security is OFF"

569: a route calculated using our client

Which / how many tiles did the client request to calculate this route?
What is the relevance of the zoom level on this process?

579: the ubiquitous FIWARE QuantumLeap API

I would not call the FIWARE QuantumLeap API ubiquitous.
It would be more usefull to compare to typical APIs that are used in the Air Quality domain and that are accepted INSPIRE best practices, such as OGC SOS or the OGC SensorThings API.

624: Figure 6

The Orion context broker is depicted twice, once inside the LTS Server, and once separately. Does this mean there are two instances of the broker running?

633: (a) the Data Event manager

This component is not listed in Figure 6, is this the second Orion context broker?

665: a subscription between the Orion context Broker and the QuantumLeap TSDB

This subscription is not visible in Figure 6.

673: Figure 7 shows that the CPU use
686: Figure. 7. Overview of the benchmark with memory cost

One of these is wrong. Which is it?

675: The memory usage of the FIWARE QL API remains stable (Figure 6)

Figure 6 is the architecture diagram.

676: At a load of four hundred clients

How many requests do those 400 clients make? One per second? One per minute? One?

Figure. 9 / Figure. 10

A scale of seconds would make the x-axis easier to read.

How many of the requests in each case are NGINX cache-hits,
how many are passed on to the back-end,
and how many result in Database requests?

What is the exact request made by the clients in both cases, and is the returned data equivalent?

How large is the data set? There are 18 vans, but how many time-series per van, and how many Observations location updates per hour, over how long a time period, resulting in how many total Observations / Locations?

701: The CPU cost ... (Figure 11)
723: Figure. 11. ... memory cost

One of these is wrong. Which is it?

716: From the second request, there is cache reuse of 30/31

I suppose the data is chunked into daily chunks? It would be nice if there was a bit more information about this.
Given this, I would expect the performance of LTS to be a factor 30 better. Is this the case? It seems the performance is a factor 60 better. What is the difference in database queries between the two setups?

751: Figure. 15

Both QL lines are solid, one should be dashed.

Figure 21 & 22

These figures (or the captions) are swapped.

Figure. 19 & Figure. 22.

The line style for the QL DB should be the same as in other figures.

808: ubiquitous QuantumLeap API

It can hardly be called ubiquitous, it is not even standardized by a standardization organization. I'd say it is rather obscure.
It would be more valuable to benchmark against an INSPIRE-sanctioned API that administrators are more likely to use for publishing time series of Observational data, such as the SensorThings API.

776: The load on the clients of the LTS Server API is 367 millicpu, compared to 2 millicpu in the case of the clients of the FIWARE QL API. As the clients of the LTS Server API need to calculate the average, their load is significantly higher.
815: Scenario b4 illustrates the balancing efforts between the publisher and the reuser. The load on the clients of the FIWARE Quantum Leap API is 185 times higher, as the clients must calculate the average sensor values.

These two paragraphs contradict each other.

836: A non-measurable benefit is that the client can evaluate any given query on the client-side, without having to rely on server-side functionality other than downloading the right fragments.

That is just as much, if not more, a disadvantage. If my query is "Give me the measurements that crossed a given threshold", then in the LDF case the client must download ALL measurements, even if no measurement matches the query.

877: We showed that is feasible to apply a LDF-based approach for publishing railway infrastructure data, which bring the same cost-efficiency and scalability benefits as the ones measured by our benchmark on air quality sensor data.

Vector tiles are not new. The advantage of Semantic vector tiles has not been shown in this paper.

880: Moreover, thanks to our data publishing approach we make possible to support route calculation use cases, that are not supported by the standard query language for semantic data SPARQL [83]

Your querying mechanism also does not support route calculation. The client did the route calculation regardless of the querying mechanism. The client can do that just as well on results from a SPARQL query, or any other vector-tile service.

882: This architectural design also allows for greater flexibility in client application design.

This architectural design forces all complexity onto the client. Which may be fine in some cases, but not in many other cases.
The fragmentation method must be carefully matched to the use-case, or the data becomes much harder to use. This aspect is completely neglected in the paper.

891: Our research indicated that the method of Linked Data can support interoperability at the technical, semantical, organizational and legal level.

Hardly. The different IOP levels are described, but that LD supports those, and how it supports them, is not well explained in this paper.

893: Our research demonstrates the significant benefits of adopting the principles of Linked Data regarding air-quality time series as these principles not only provide interoperability towards external stakeholders but also foster a more sustainable and cost-effective architecture.

The benefits of adopting linked data are not shown, only referenced. This paper shows that static data can be served with fewer resources than dynamic data. But this applies just as well for CSV files as for LD Fragments.

The paper mixes two topics, using Linked Data Fragments for Air Quality & Route data, and performance advantages of serving static Fragments. Unfortunately, the former is not well covered, and that serving static data is faster than serving dynamic data from a database has been known for 20 years.

Reviewer 2 Report

The paper focuses on how data providers can develop a sustainable method for publishing open sensor data, in particular time series of air quality sensor data. The experiment set up showed a reduction in the cost of publishing air quality data and an increase in availability through an improved caching strategy. The paper is well written and organised. Nevertheless, some timely suggestions follow that need to be implemented to improve the work:
- the text from line 430 to line 440 is repeated twice.
- on line 499 it is said that the ERA vocabulary used as a knowledge graph has already been introduced in section 2.2; this is not true.
Furthermore, the paper needs to be enriched with the following insights:
- knowledge graphs: it must be clarified what they are, how they differ from ontologies, which and how they are used, with which tools.
- semantic interoperability: it should be better clarified, perhaps with practical examples, how to overcome the obstacle of the lack of semantic interoperability between data, vocabularies, even pre-existing sources that manage the same concepts ... this is the holy grail that allows not to waste the huge amount of data already present in multiple domains.
- bibliography: I suggest supplementing the bibliography with some more general papers to contextualise the work in the context of smart cities, for example by mentioning the need to publish interoperable data on sustainable energy, water supply networks management, policymaking, for example:
- Escobar, P., Roldán-García, M. D. M., Peral, J., Candela, G., & García-Nieto, J. (2020). An Ontology-Based Framework for Publishing and Exploiting Linked Open Data: A Use Case on Water Resources Management. Applied Sciences, 10(3), 779.
- Carbonaro, A. (2019). Linked data and semantic web technologies to model context information for policy-making. Journal of Ambient Intelligence and Humanized Computing, 1-12.
- Anthony Jnr, B., Abbas Petersen, S., Ahlers, D., & Krogstie, J. (2020). API deployment for big data management towards sustainable energy prosumption in smart cities-a layered architecture perspective. International Journal of Sustainable Energy, 39(3), 263-289.

Reviewer 3 Report

I’m not sure the term “Web Scale” as mentioned in the title is the most appropriate.

First of all, I suggest to clarify in the introduction aims and scope. It’s not that clear to me.

Section 2 needs a proper introduction. I suggest also to review the terminology as “interoperability applied to X” sounds really forced. I suggest a re-structure of the section to first clearly discuss problems related to interoperability and then to discuss peculiarities within the different domains, as well as cross-domain issues. The current structure is honestly not that effective.

The current technological context (e.g. the Semantic Web) should be addressed in generic terms.

Also section 3 misses a proper introduction. It mentions semi-structured interviews. I strongly suggest a paragraph or short section in the introductory part to explain the methodology adopted for the research conducted.

Similar considerations apply to all other parts as the paper doesn’t propose a fluid discussion but rather jumps from a topic to another.

It would be useful to try to understand urban data sharing at a more holistic level, for instance looking at the different applications  (e.g. urban planning) and novel metrics (e.g. journey-to-work data). Some of the very many works available:

  • Shen, Jian, et al. "A secure cloud-assisted urban data sharing framework for ubiquitous-cities." Pervasive and mobile Computing 41 (2017): 219-230.
  • Moere, Andrew Vande, and Dan Hill. "Designing for the situated and public visualization of urban data." Journal of Urban Technology 19.2 (2012): 25-46.
  • Guohun Zhu, Jonathan Corcoran, Paul Shyy, Salvatore Flavio Pileggi, Jane Hunter, Analysing journey-to-work data using complex networks, Journal of Transport Geography, Volume 66, 2018, Pages 65-79.
  • Barbosa, Luciano, et al. "Structured open urban data: understanding the landscape." Big data 2.3 (2014): 144-154.
  • Reades, Jonathan, et al. "Cellular census: Explorations in urban data collection." IEEE Pervasive computing 6.3 (2007): 30-38.

Overall, the paper could be of interest for readers but MUST be re-arranged and improved in terms of both content and readability.

Round 2

Reviewer 1 Report

The changes and extra clarifications make the paper much easier to understand and easier to follow.

I did find one change that seems to have introduced a contradiction:

Table 2 states there is a Ratified vocabulary for airquality, linking to a github page.
Line 284-286 states: At the time of writing and to the best of our knowledge, no ontologies for air quality were ratified by a standardisation organisation.

Reviewer 3 Report

I believe the paper has been significantly improved.

I would suggest just to additionally extend the explanations provided as per previous round comments.

Round 3

Reviewer 1 Report

No further comments.

Author Response

Thank you for your feedback.